# *Helicobacter pylori* FabX contains a [4Fe-4S] cluster essential for unsaturated fatty acid synthesis

Jiashen Zhou [1,11], Lin Zhang[1,11], Liping Zeng[2,11], Lu Yu [3,11], Yuanyuan Duan[4,11], Siqi Shen[1,11], Jingyan Hu[1], Pan Zhang[5], Wenyan Song[1], Xiaoxue Ruan[6], Jing Jiang[7], Yinan Zhang [7], Lu Zhou[6], Jia Jia[2], Xudong Hang[2], Changlin Tian[3,8], Houwen Lin[1], Hong-Zhuan Chen[9✉], John E. Cronan [10✉], Hongkai Bi[2✉] & Liang Zhang [1✉]

Unsaturated fatty acids (UFAs) are essential for functional membrane phospholipids in most bacteria. The bifunctional dehydrogenase/isomerase FabX is an essential UFA biosynthesis enzyme in the widespread human pathogen *Helicobacter pylori*, a bacterium etiologically related to 95% of gastric cancers. Here, we present the crystal structures of FabX alone and in complexes with an octanoyl-acyl carrier protein (ACP) substrate or with holo-ACP. FabX belongs to the nitronate monooxygenase (NMO) flavoprotein family but contains an atypical [4Fe-4S] cluster absent in all other family members characterized to date. FabX binds ACP via its positively charged α7 helix that interacts with the negatively charged α2 and α3 helices of ACP. We demonstrate that the [4Fe-4S] cluster potentiates FMN oxidation during dehydrogenase catalysis, generating superoxide from an oxygen molecule that is locked in an oxyanion hole between the FMN and the active site residue His182. Both the [4Fe-4S] and FMN cofactors are essential for UFA synthesis, and the superoxide is subsequently excreted by *H. pylori* as a major resource of peroxide which may contribute to its pathogenic function in the corrosion of gastric mucosa.

---

[1] Department of Pharmacology and Chemical Biology, State Key Laboratory of Oncogenes and Related Genes, Shanghai Jiao Tong University School of Medicine, 200025 Shanghai, China. [2] Department of Pathogen Biology & Jiangsu Key Laboratory of Pathogen Biology & Sir Run Run Hospital, Nanjing Medical University, 211166 Nanjing, Jiangsu, China. [3] High Magnetic Field Laboratory, Chinese Academy of Sciences, 230031 Hefei, China. [4] The Laboratory Center for Basic Medical Sciences, Nanjing Medical University, 211166 Nanjing, Jiangsu, China. [5] Key Laboratory of Medical Epigenetics and Metabolism, Institutes of Biomedical Sciences, Shanghai Medical College, Fudan University, 200032 Shanghai, China. [6] Department of Medicinal Chemistry, School of Pharmacy, Fudan University, 201203 Shanghai, China. [7] Jiangsu Key Laboratory for Functional Substance of Chinese Medicine, School of Pharmacy, Nanjing University of Chinese Medicine, 210023 Nanjing, Jiangsu, China. [8] The First Affiliated Hospital of USTC, Division of Life Sciences and Medicine, and Center for BioAnalytical Chemistry, Hefei National Laboratory of Physical Science at Microscale, University of Science and Technology of China, 230026 Hefei, Anhui, China. [9] Institute of Interdisciplinary Integrative Biomedical Research, Shuguang Hospital, Shanghai University of Traditional Chinese Medicine, 201203 Shanghai, China. [10] Departments of Microbiology and Biochemistry, University of Illinois, Urbana, IL 61801, USA. [11] These authors contributes equally: Jiashen Zhou, Lin Zhang, Liping Zeng, Lu Yu, Yuanyuan Duan, Siqi Shen. ✉email: hongzhuan_chen@hotmail.com; jecronan@illinois.edu; hkbi@njmu.edu.cn; liangzhang2014@sjtu.edu.cn

Fatty acids are essential for functional cell membrane lipids. The ratios of unsaturated (UFA) to saturated fatty acids (SFA) adjust membrane fluidity to allow function in changing environments[1,2]. In bacteria, SFA and UFA are synthesized by a highly conserved type-II biosynthesis system (FAS-II) in which a series of discrete enzymes recognize the ACP-bound substrates (ACP) and assemble the acyl chains in a step-by-step ordered cycle (Fig. 1a)[3,4].

SFA biosynthesis is conveniently divided into two stages, initiation and cyclic elongation. ACP is post-translationally modified by attachment of a phosphopantetheine group (PPant) derived from CoA to give holo-ACP, the active form of the protein. The acyl chain intermediates in the pathway are covalently attached to ACP via thioester linkages to the thiol of the PPant. The key building block of fatty acids is malonyl-ACP which adds two carbons to each acyl-ACP intermediate via a decarboxylative Claisen condensation. Enzymes catalyzing ketoreduction, dehydration, and enoyl-reduction biochemical catalysis then prepare the chain for the next decarboxylative Claisen condensation. When the acyl chain is sufficiently long

(16–18 carbons), the acyl-ACP becomes a substrate for incorporation into phospholipids. The enzymes of SFA biosynthesis have been well studied[3,4] and one of the native enzyme/holo–ACP complex structure has been solved[5,6], as well as several covalently crosslinked enzyme–ACP structures[7–11].

In contrast, UFA synthesis utilizes the catalytic steps that are the same as those of SFA synthesis, excepting the introduction of the *cis*-double bond into the acyl chain. The *cis*-double bond is introduced by two different pathways in bacteria, desaturation of the full length SFA or by insertion of a *cis*-double bond into a short chain (8–10 carbons) acyl-ACP intermediate. Following the introduction of the *cis*-double bond, the nascent UFA chains are elongated to the 16–18 carbon lengths required for incorporation into phospholipids. Desaturation is limited by the requirement for molecular oxygen whereas the insertion–elongation pathway (generally called the anaerobic pathway) functions irrespective of oxygen. The anaerobic pathways generally proceed together with or after the dehydration step of the elongation cycle by utilizing either a dehydratase/isomerase bifunctional enzyme such as *Escherichia coli* FabA (in most Gram-negative bacteria)[7,12], FabN

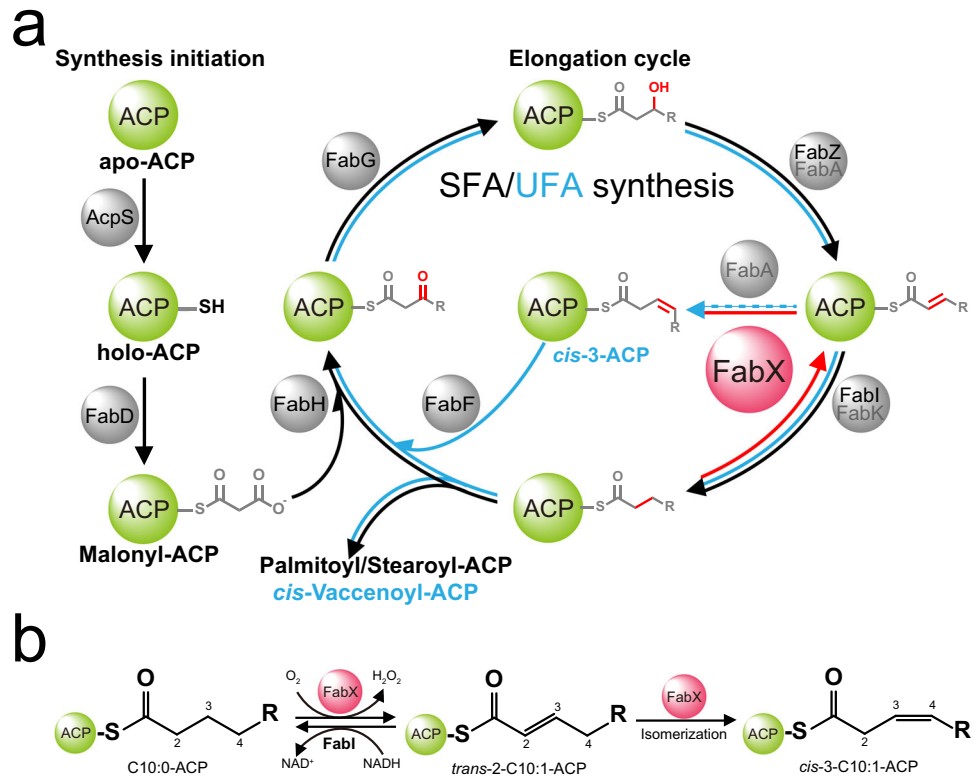

**Fig. 1 Schematic diagram of fatty acid biosynthesis in *H. pylori*. a** The SFA (black arrows) and UFA (blue arrows) biosynthesis pathways share the same initiation steps, but have distinct differences in the elongation cycle. In the initiation stage, apo form ACP (apo-ACP) is firstly activated to holo form ACP (holo-ACP) by acyl carrier protein synthase (AcpS), where the key residue Serine (Ser36 in *H. pylori*) of apo-ACP is covalently attached by a phosphopantetheine group (PPant) derived from CoA. Malonyl-CoA:acyl carrier protein transacylase (FabD) transfers the initial three-carbon substrate to the thiol group of the PPant arm on holo-ACP. Subsequently, the Malonyl-ACP moves to the elongation cycle. For SFA synthesis, the acyl-ACP intermediate is catalyzed in the order of decarboxylative Claisen condensation (by FabH) → ketoreduction (by FabG) → dehydration (by FabZ and FabA, while *H. pylori* lacks a *fabA* gene) → enoyl-reduction (by FabI and FabK, while *H. pylori* lacks a *fabK* gene). After these steps, two carbons are added to the acyl-ACP intermediate, and the saturated intermediate move to the next cycle for further elongation until it reaches sufficiently long (16–18 carbons) and releases. For UFA synthesis, the *cis*-double bond is introduced after the dehydration step in the elongation cycle by bifunctional dehydratase/isomerase FabA in most Gram-negative bacteria. FabA catalyzes the dehydration of 3-hydroxydecenoyl-ACP intermediate (<12 carbons), and isomerizes the *trans*-2-product to the *cis*-3-product. The *cis*-3-ACP is further catalyzed for decarboxylative Claisen condensation (by FabB and FabF, while *H. pylori* lacks a *fabB* gene), and the product moves to the elongation cycle again. Due to the lack of *fabA* gene in *H. pylori* genome, we discovered that *H. pylori* utilizes FabX to introduce *cis*-double bond. As shown in the figure, FabX catalyzes the dehydrogenative reaction of saturated acyl-ACP intermediate, as well as the subsequent isomerization reaction. FabX, and the dehydrogenative/isomerization reactions it catalyzes are colored in red. **b** Schematic diagram of FabX dehydrogenative/isomerization catalysis, and the FabX–FabI coupled assay used in the kinetic characterizations.

(in *Enterococcus faecalis* and *Tetracoccus salidis*)[13], *Aerococcus viridans* FabQ[14], or a mono-functional isomerase, such as *Streptococcus pneumoniae* FabM[15] to generate the *cis*-double bond in all cases followed by acyl chain elongation[10,11,16,17].

In 2016, our group discovered that the human pathogen, *Helicobacter pylori* utilizes a remarkably atypical enzyme, FabX, to introduce the *cis*-double bond[18]. FabX is a bifunctional dehydrogenase/isomerase enzyme, that introduces the UFA *cis*-double bond via an backtracking mechanism more akin to fatty acid degradation than fatty acid synthesis. The oxygen-requiring dehydrogenation reaction generates a *trans*-2-enoyl-ACP that is isomerized to the *cis*-3-decenoyl-ACP required for UFA synthesis. FabX reverses the normal direction of the fatty acid synthesis cycle by catalyzing dehydrogenation of the saturated acyl chain of decanoyl-ACP, in a chemically difficult reaction opposite that of the last step (enoyl-reduction) of the classical elongation cycle (Fig. 1b)[18]. Genes that encode putative FabX enzymes are rare in bacteria associated with humans, and thus inhibitors targeting these enzymes should be quite specific for a given pathogen without significant disruption of healthy gut microbiota.

FabX belongs to the nitronate monooxygenase (NMO) flavoprotein family, where the conventional enoyl-ACP reductase FabKs also belong to. FabKs consist of a family conserved TIM-barrel fold structure that holds a FMN cofactor, and a NAD(P)H-binding domain. They utilize the key histidine located next to the FMN for nucleophilic attack of the hydrogen atom on the substrate and thereby initiates the catalytic reaction, and NAD(P)H cofactor to provide electrons to achieve the subsequent reductase activity[19–21]. Moreover, other studies on the NMO family member L-lactate oxidase (LOX) showed an oxygen molecule bound next to the sidechain of the key histidine, and functions as the electron acceptor[22]. Although FabX was shown to be a flavoenzyme that has an FMN cofactor and catalyzes an oxygen requiring reaction, the mechanism of the catalytic cycle was obscure[18].

In this work, we obtain the crystal structures of FabX, a FabX–octanoyl-ACP complex and a FabX–holo-ACP complex, and reveal a dynamic acyl-ACP recognition mechanism of FabX. Moreover, the enzyme is found to contain a second electron transfer component, a [4Fe–4S] cluster, in addition to the FMN cofactor, which to date has not been observed in enzymes involved in the fatty acid synthesis. We demonstrate that the [4Fe–4S] cluster functions as a temporary electron depot that receives electrons from FMN and sends electrons back to FMN to achieve independent bidirectional internal electron transfer. Lastly, we demonstrate that the superoxide generated by FabX catalysis as a major source of peroxide excreted by *H. pylori* which may contribute to its pathogenesis by corrosion of gastric mucosa. These findings may facilitate structure-guided development of antibacterials targeting *H. pylori*.

## Results

**Biological importance of FabX**. In the prior paper we argued that FabX is essential for growth of *H. pylori*[18]. This argument was based on the strict conservation of FabX in very diverse *H. pylori* strains and that over 5000 transposon insertions had been mapped in *H. pylori* genomes without finding an insertion into the *fabX* gene. To support these inferences, we attempted to directly delete the *H. pylori fabX* gene[18]. However, no mutant strains were obtained in multiple transformation experiments even when a generic UFA was included in the medium. To provide a means to allow survival of *H. pylori* strains lacking the *fabX* gene, we introduced a synthetic copy of the *E. coli fabA* gene driven by a *H. pylori* promoter. The rationale was the

dehydrogenation activity was only an unusual means to provide the *trans*-2-decenoyl-ACP for isomerization to the essential *cis*-3-decenoyl-ACP. Hence, if FabA could access 3-hydroxydecenoyl-ACP in *H. pylori*, an intermediate in SFA synthesis, then the dehydratase-isomerase activity of the bifunctional *E. coli* FabA enzyme should allow loss of *fabX* to be bypassed. FabA would dehydrate 3-hydroxydecenoyl-ACP to *trans*-2-decenoyl-ACP and isomerize it to *cis*-3-decenoyl-ACP. As expected, expression of *E. coli* FabA allowed the *H. pylori fabX* gene to be readily deleted (Fig. S1) and the replacement of FabX by FabA did not affect the growth of *H. pylori* (Fig. S2). Hence, FabA could access the 3-hydroxydecenoyl-ACP intermediate of the FAS-II pathway in *H. pylori* and produce the *cis*-3-decenoyl-ACP needed for UFA synthesis. This raises the question of why *H. pylori* uses the chemically difficult dehydrogenation reaction to generate *trans*-2-decenoyl-ACP when given a FabA paralog, the conventional pathway would suffice. Our inability to isolate *fabX* deletions using media supplemented with UFA may be because *H. pylori* has not developed the capacity to incorporate UFA because free UFAs are not abundant in the stomach. Free UFAs are produced from food stuffs only upon hydrolysis of complex lipids (e.g., triglycerides and phospholipids) which occurs further down in the digestive system (the small intestine).

**Substrate specificity of FabX**. In order to understand the biological function of FabX, we determined the dependence of the FabX dehydrogenation activity against ACP thioesters of various acyl chain lengths (hexanoyl-, octanoyl-, decanoyl- or dodecanoyl-) by coupling the reaction to the *E. coli* enoyl-ACP reductase FabI reaction and measuring the decrease of NADH absorbance at 340 nm (Fig. 1b)[18,23]. The results indicate that FabX prefers decanoyl-ACP and octanoyl-ACP rather than shorter or longer acyl-ACP substrates (Fig. S3). Moreover, the kinetic efficiency of FabX catalyzing decanoyl-ACP dehydrogenation is ~two-fold faster than that of octanoyl-ACP and nearly a 1000-fold faster than seen with the substrate analog, decanoyl-CoA (Figs. 2a, S4 and Table S1). Hence, FabX prefers C10 and C8-acyl chains carried by ACP.

**The overall crystal structure of FabX**. We solved the crystal structure of FabX at 1.7 Å resolution with a single FabX monomer observed in the asymmetric unit (Table S2). The full length of FabX contains 363 amino acid residues, all of which were observed in the structure. The overall structure of FabX predominantly consists of a large canonical TIM-barrel fold subdomain at its N-terminus (residues from 1 to 266, including β1–β8 and α1–α11) (Fig. 2b). It contains a FMN cofactor-binding core that consists of eight continuous β-strand–α-helix repeats (β3–α5–β4–α6–β5–α7–β6–α8), and a FMN molecule is non-covalently bound inside the core (Fig. S5). The TIM-barrel fold, the FMN core as well as the essential NMO catalytic histidine residue of FabX (H182) are highly conserved in FabK homologs at both the amino acid sequence and tertiary structural levels (Figs. S6 and S7)[19–21].

The FabX FMN cofactor is held at an appropriate place inside the active pocket by hydrophilic interactions with the sidechains of N99, E175, Q240, and T243, and the backbones of G25, G180, G221, A242, and T243, as well as hydrophobic interactions with G22, G23, M24, I28, L101, I147, E175, S179, G220, M241, F339, T340, and G341 (Fig. S8). In contrast, the catalytic residue H182 is located ~6.5 Å from the FMN cofactor, and is stabilized through hydrogen bonding with the sidechain of S273 (Fig. S8). Previous studies on NMO family enzymes such as FabKs suggested that such catalytic histidine functions as a "reaction trigger" as the $N^{\delta1}$ atom of the sidechain of the histidine residue

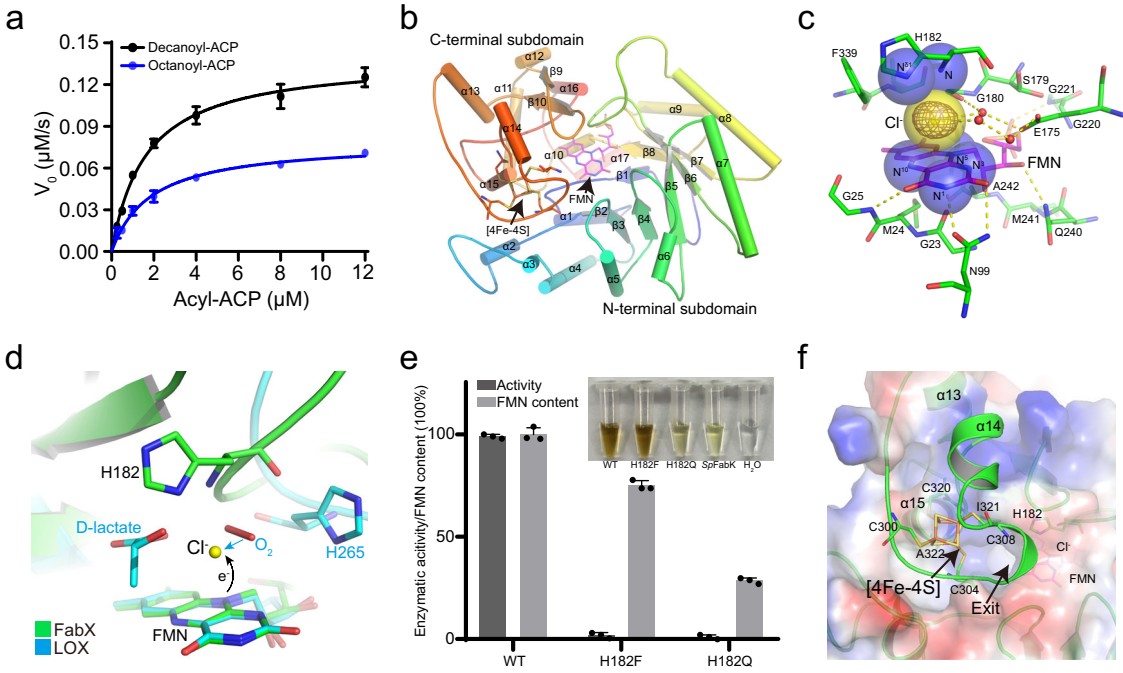

**Fig. 2 Structural characterization of FabX. a** Kinetic characterization of FabX catalysis with decanoyl-ACP (black) and octanoyl-ACP (blue). The curves were fitted to the Michaelis–Menten equation. $N = 3$ biologically independent experiments. Data is presented as mean ± SD. **b** The overall crystal structure of FabXs colored blue to red from N- to C-terminus. The FMN cofactor and [4Fe–4S] cluster are shown as sticks. The two subdomains, secondary structural elements, FMN cofactor, and [4Fe–4S] cluster are labeled. **c** The FMN-binding motif of FabX. The residues and water molecules involved in FMN binding are shown in green sticks and red spheres, respectively. Yellow dashes indicate hydrogen bonds between FMN and surrounding amino acid residues. The nitrogen atoms (N and $N^{\delta 1}$) of H182 and FMN ($N^1$, $N^3$, $N^5$ and $N^{10}$) that contribute to oxyanion hole formation are presented as labeled transparent blue spheres. The chloride ion found locked in the oxyanion hole is shown as a transparent yellow sphere, and its fofc electron map ($\sigma = 3$) is shown in brown mesh. **d** Structural superposition of FMN-binding motif of FabX (green) with that of lactate oxidase (LOX, cyan, pdb code: 2NLI)[23]. The FMN cofactor, key histidine residue (His265), and D-lactate of the LOX structure are shown as cyan sticks, while the $O_2$ molecule observed in the LOX structure is shown in red sticks and labeled. FabX His182, FMN and the chloride ion are shown in green sticks and yellow sphere, respectively. The cyan arrow indicates that the replacement of the chloride ion with an oxygen molecule during catalysis. **e** Dehydrogenation activities (black) and FMN contents (gray) of FabX, FabX H182F and FabX H182Q variants, determined by the FabX–FabI coupled assay and LC–MS/MS, respectively. The protein color of the variants was shown and labeled. $N = 3$ biologically independent experiments. Data is presented as mean ± SD. **f** The electrostatic surface of FabX with the [4Fe–4S] cluster binding residues. Four key cysteine residues (C300, C304, C308, and C320) and the residues involved in interactions with the cluster are shown as sticks and labeled. The exit of the active site tunnel of [4Fe–4S] cluster is indicated by black arrows. Source data are provided as a Source Data file.

abstracts a proton from the substrate through nucleophilic attack to initiate the reaction and to facilitate the subsequent electron transfer on FMN[19–21].

Moreover, the H182 nitrogen atoms of the sidechain ($N^{\delta 1}$) and backbone (N) together with four nitrogen atoms ($N^1$, $N^3$, $N^5$, and $N^{10}$) of FMN form a positive charged oxyanion hole located between the H182 sidechain and the FMN cofactor, which has never been observed in this family (Fig. 2c)[24,25]. A chloride anion was found locked in the oxyanion hole and is stabilized through hydrogen bonds with E175 and G180 mediated by nearby water chains. Superposition of the FabX structure with the structure of LOX in complex with D-lactate and oxygen (pdb code: 2NIL) shows that the position of oxygen in the LOX structure is very close to the oxyanion hole of FabX (Fig. 2d). This superposition argues strongly that the oxyanion hole of FabX is able to hold an oxygen molecule to act as potential electron acceptor for reoxidation of the FMN cofactor during catalysis[22].

As expected, mutagenesis of H182 to phenylalanine (FabX H182F) abolished the FabX dehydrogenation ability without significantly interfering with FMN binding (the FMN content remains ~80%) or protein stability (Figs. 2e and S9–S11). Moreover, the hydrophobic imidazole ring of H182 sidechain is required for orienting FMN by hydrophobic interactions.

Substitution of H182 with the nearly isosteric glutamine (FabX H182Q) resulted in a sharp decrease of the FMN content to ~28%, reduced protein stability by 7.2 °C, and completely abolished enzymatic activity (Figs. 2e, S10 and S11). These observations argue that the imidazole ring of H182 is required for maintaining both protein stability and for the FMN cofactor to achieve dehydrogenation activity due to π stacking.

In addition to the canonical TIM-barrel fold core, FabX was unexpectedly found to contain a small [4Fe–4S] cluster-binding subdomain at its C-terminus (residues from 267 to 363), which has never been observed in enzymes involved in the fatty acid synthesis (Figs. 2b and S12). The subdomain consists of six α-helices and two β-strands (α12–α17 and β9–β10), and four cysteine residues C300, C304, C308 and C320 hold the [4Fe–4S] cluster, a second electron transfer component other than FMN, through coordination bonding interactions (Figs. 2f and S12). The [4Fe–4S] cluster is stabilized through hydrophobic interactions with residues located on the N-terminus of α15, such as Ile321 and Ala 322, and blocks exit from the active site tunnel (Figs. S12 and S13). Interestingly, although FabX and FabKs share the highly conserved TIM-barrel fold structure as well as the key catalytic histidine residue, they catalyze opposite reactions (dehydrogenation vs. reduction) (Fig. 1). The structural

comparison between FabX and FabK from *Porphyromonas gingivalis* (*Pg*FabK, PDB code: 4IQL, *H. pylori* lacks a *fabK* gene) suggests that the [4Fe–4S]-binding domain in FabX replaces the NAD(P)H-binding motif of *Pg*FabK (Fig. S7). This raises the possibility that the opposing reactions of FabX and FabKs could be due to the FabX [4Fe–4S] cluster binding domain and the FabKs NAD(P)H-binding domain.

To elucidate the function of the [4Fe–4S] cluster in catalysis, the four cysteine residues (C300, C304, C308, C320) that structure the [4Fe–4S] cluster were mutated to serine or alanine. Interestingly, all mutations except C304S and C304A completely destroyed the cluster (Fig. S14a and b), destabilized the protein (Fig. S14c) leading to the loss of the FMN cofactor (Fig. S14a), and abolished the enzymatic activity (Figs. S14a and S15). In contrast, an appreciable fraction of the C304S and C304A proteins partially retained the cluster and ~80% of the FMN content (Figs. S14d and S16). This could be explained by that the fact C304 is the most buried ligand inside the active site cavity (Fig. S12). The replacement of C304 by an alanine would lead to a cluster with a ligand exchangeable site, and prevents losing the iron and thus increases cluster stability. However, the catalytic efficiency of C304A is ~20-fold lower than that of wild type FabX, indicating that the [4Fe–4S] cluster is required for the FabX catalysis (Fig. S14e and Table S1).

In the overall structure of FabX, the FMN core from the large subdomain and the [4Fe–4S] cluster from the small subdomain are linked by a "L" shaped hydrophobic catalytic tunnel (Fig. S13). The FMN core is located near the kink of the tunnel, whereas the [4Fe–4S] cluster is located at the end of the tunnel where it blocks the end, and is 13.4 Å away from the FMN cofactor (nearest atoms distance), which is within the theoretical distance limitation (<14 Å) that electrons can directly transfer[26]. These indicate that both of the FMN cofactor and the [4Fe–4S] cluster function in the catalytic reaction of FabX.

**Interaction of FabX with ACP**. To investigate the acyl-ACP substrate recognition mechanism of FabX, we performed crystal screening of FabX–holo-ACP and FabX–octanoyl-ACP complexes under anaerobic condition which block dehydrogenation of the substrate (FabX requires oxygen for activity)[18]. We obtained crystal structures of FabX–holo-ACP complex at 2.8 Å resolution with space group P2₁ and FabX–octanoyl-ACP complex at 2.3 Å resolution with space group P2₁2₁2₁ (Table S2 and Fig. S17). The determined structures showed that there are two FabX–ACP complex molecules in the asymmetric unit with a 1:1 binding ratio, which is in consistent with static light scattering assay results (Figs. S18 and S19).

In the FabX–octanoyl-ACP structure, one of the complexes in the asymmetric unit is well determined, while the ACP in the other complex only shows the key α2 helix (Fig. S18a). The remaining ACP segments are disordered in the structure due to discontinuous electron density. However, the two complexes in the asymmetric unit are highly identical with RMSD value of 0.4737. As shown in Figs. 3a and S20a, octanoyl-ACP binds FabX predominantly via electrostatic and hydrophobic interactions between its highly conserved α2-helix and the α7-helix of FabX. Specifically, L37′ and V40′ of the N-terminus of ACP α2-helix forms hydrophobic interactions with I156 and K152 of the N-terminus of the FabX α7-helix (the prime indicates that an ACP residue). M44′ and E47′ of the C-terminus of ACP α2-helix induce significant sidechain rotations of R160 and R164 located on the FabX α7-helix C-terminus through forming salt-bridge interactions between E47′ and R160. Moreover, D56′ from the α3-helix of ACP forms additional salt-bridges with

R164 of FabX. D56′ is hydrogen bonded to T130 on the FabX loop between α6 and β4 and has hydrophobic interactions with the sidechain of N131 (Fig. S21). These interactions ensure the precise orientation of the α2-helix such that the PPant moiety and attached acyl chain of the substrate are poised to insert into the active site tunnel. In addition, the sidechains of V40′ and M44′ from the center of ACP α2-helix form hydrophobic interactions with the sidechains of FabX I156 and K159 which further stabilize ACP–FabX binding.

Notably, the key salt-bridge interactions between the negatively charged residues D35′ and D38′ of the ACP α2-helix N-terminus and positively charged residues near the active site tunnel of FAS-II enzymes (K152 in FabX case) seen in previous acyl-ACP enzyme complexes[5,6,25] were not observed in the FabX–octanoyl-ACP complex. The sidechains of ACP D35′ and D38′ are ~5.6 and 4.4 Å away from FabX K152 (Fig. 3a). Instead, D56′ of ACP α3-helix interacts with T130 and R164 of FabX. Such α3-helix ACP interactions are rarely observed, as the previous studies shows that the enzyme–ACP interactions mainly occur on ACP α2-helix[27].

In contrast to the FabX–octanoyl-ACP complex, both of two FabX–holo-ACP complexes in the asymmetric unit were well determined and essentially identical (RMSD value of 0.4687) (Fig. S18b). holo-ACP binds FabX predominantly through its α2-helix through electrostatic and hydrophobic interactions as expected (Figs. S20b and S22). The PPant arm of holo-ACP stretches into the active tunnel of FabX, where it is stabilized through hydrophobic interactions with FabX residues including I147, S149, G184, and Y277 (Fig. S23). Moreover, the O40′ atom of the PPant arm forms an additional H-bond with the sidechain of E175. However, superposition of the FabX–holo-ACP complex with the FabX–octanoyl-ACP complex showed a surprising ~30° rotation between the ACP molecules of the two complexes (Fig. 3b). Specifically, holo-ACP rotates in the clockwise direction along the α2-helix with the sidechain of FabX R164 acting as the rotation fulcrum, allowing D35′ and D38′ from holo–ACP to form salt-bridges with the sidechain of FabX K152 (Fig. S22), which were not observed in the FabX–octanoyl-ACP complex. Moreover, the rotation weakens the interactions between α3-helix of octanoyl-ACP and FabX, and abolishes the interaction between D56′ of octanoyl-ACP and FabX T130 (Fig. 3a and b). In contrast, in the FabX–octanoyl-ACP complex, the sidechain of D56′ of ACP α3-helix inserts into the chink between helices α6 and α7 of FabX (Fig. 3b). This pushes away the sidechain of N131 located on the loop between α6 and β4, and forms additional hydrophilic interactions with FabX R164 and T130.

Notably, such dynamic conformational changes observed upon binding of octanoyl-ACP compared to holo–ACP binding have not been reported previously. To confirm that the observed conformational changes were not due to the crystal packing forces, FabX R164 to alanine (FabX R164A) mutagenesis was performed. Disruption of the interactions by mutating FabX R164 to alanine (FabX R164A) abolished acyl-ACP binding (Fig. 3c) and reduced in vitro dehydrogenation activity by ~80 fold (Fig. 3d). The mutation also significantly retarded *H. pylori* growth in vivo (Figs. 3e and S24, Table S1). These observations suggested that the interactions between α3-helix of ACP (D56') and the FabX chink (R164 and T130) plays a unique and pivotal role in FabX substrate recognition and catalysis.

Moreover, sequence alignments of FabX homologs show that the key residues of FabX involved in the interactions with ACP are conserved in FabX homologs, but not in the FabK family (Figs. S6 and S25). These results indicate that FabX employs an atypical acyl-ACP recognition mechanism compared to those of FabKs.

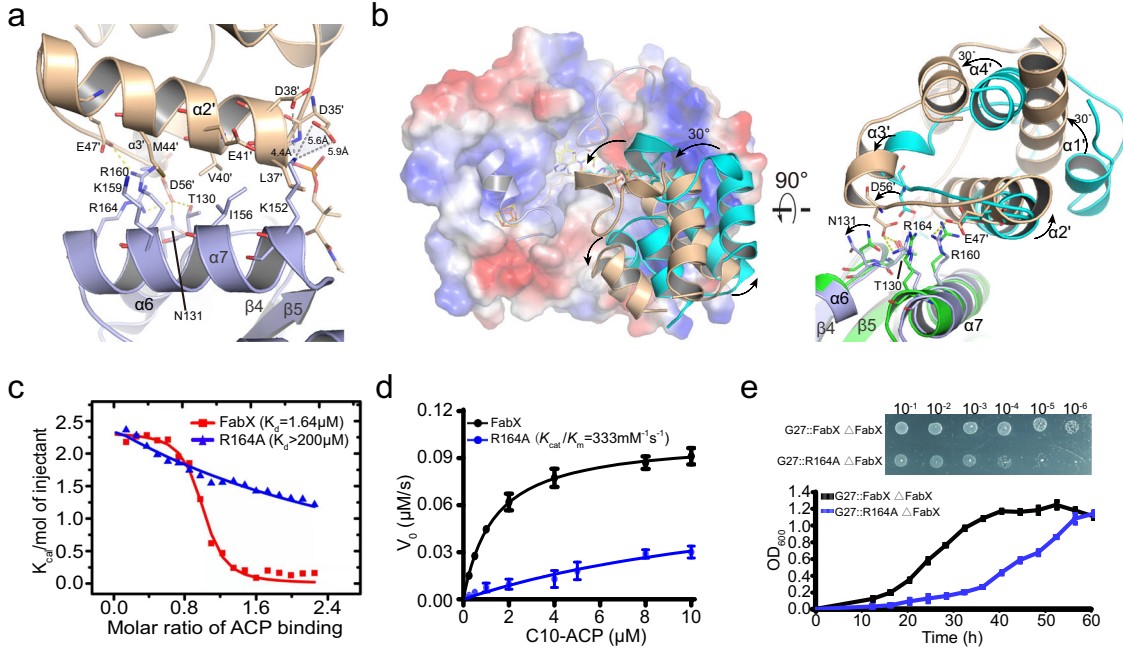

**Fig. 3 Interactions between FabX and ACP. a** Interactions between octanoyl-ACP (wheat) and FabX (light blue). Residues involved in the binding are shown in sticks and labeled. The primes indicate an ACP residue. Yellow dashes indicate hydrogen bonds between FabX and ACP residues. Gray dashes indicate the absence of hydrogen bonds that are normally observed in ACP–enzyme complexes of FAS-II pathways, and the distances are labeled. **b** Superposition of FabX–holo-ACP complex (green/cyan) with FabX–octanoyl-ACP complex (light blue/wheat). The electrostatic surface of FabX bound by octanoyl-ACP (wheat) or holo-ACP (cyan) is shown. Half of the active site tunnel inner surface and the secondary structures of [4Fe–4S] cluster are shown. The black arrows indicate the rotation direction of the secondary elements or the sidechains of residues between the two complexes. The FMN cofactor and H182 were also shown in sticks and labeled. **c** Isothermal titration calorimetry (ITC) determination of holo-ACP binding to FabX (red) or FabX R164A variant (blue). **d** Kinetic characterization of FabX (black) or FabX R164A (blue) dehydrogenase activity in catalyzing decanoyl-ACP. $N = 3$ biologically independent experiments. Data is presented as mean ± SD. **e** The growth phenotypes of the strains BHKS487 (G27 IR0203::*fabX* Δ*fabX*) and BHKS488 (G27 IR0203::*fabX R164A* Δ*fabX*). The strains were cultured in BHI medium containing 10% FCS and Columbia blood agar plates containing 5% FCS, respectively. $N = 3$ biologically independent experiments. Data is presented as mean ± SD. Source data are provided as a Source Data file.

**Mechanism of FabX recognition of the acyl substrate carried by ACP.** The anticlockwise rotation of ACP on the α7-helix of FabX poises the octanoyl chain plus the thioester proximal portion of the PPant prosthetic group to insert into the "L" shaped hydrophobic tunnel of FabX through the oxyanion hole between FMN and H182. The methyl end of the acyl chain points towards the [4Fe–4S] cluster (Fig. 4).

Compared to the PPant conformation in the holo-ACP structure in the active site tunnel, the octanoyl-PPant is rotated ~60° relative to the ACP rotation, and is stabilized through hydrophobic and hydrophilic interactions with residues in the tunnel (Fig. 4). In the FabX–octanoyl-ACP complex, the PPant is stabilized predominantly through hydrophobic interactions with the FabX sidechains of S149, I147, Y277 plus the backbones of G127 and G184. Moreover, rotation of the PPant moiety inside the tunnel introduces several additional H-bonds in addition to the original hydrophilic interaction between O40′ atom and the sidechain of E175 that was observed in FabX–holo-ACP structure. These include the interactions between O33′, O35′, and O40′ from the ACP PPant moiety with the backbones of FabX S149 and G127, and the sidechain of E175, respectively. These interactions firmly orient the substrate for catalysis.

In addition to the PPant–FabX interactions, the octanoyl chain attached to PPant is stabilized in the tunnel through hydrophobic interactions with surrounding residues including G25, V26, V49, G50, F74, Y75, L101, H182, V305, I321, L325, and F339. The oxygen atom (O41′) of the octanoyl thioester forms hydrophilic interactions with the sidechain of E175 and the backbone of H182 via a water molecule, ensuring that the acyl substrate adopts the

required conformation. The methyl end of the octanoyl chain almost encloses the [4Fe–4S] cluster, indicating that there would be insufficient space in current FabX conformation to accommodate of the two additional carbons of a decanoyl chain, despite decanoyl-ACP being the best FabX substrate (Fig. S26a). The docking of a decanoyl group to FabX further confirmed this observation (Fig. S26b and c). The decanoyl chain adopts a similar conformation compared to that of the octanoyl chain. However, the methyl end of the decanoyl chain shows strong clashes with the [4Fe–4S] cluster and residues around. These results indicate that there must be some conformational change induced during decanoyl-ACP binding for accommodation of the decanoyl substrate. Notley, previous studies demonstrated that FabA, the dehydratase/isomerase that catalyzes unsaturated fatty acid biosynthesis in most of the bacteria, performs dehydration of C8 and C12 substrates about 80% as well as C10 in vitro[28]. Moreover, we previously demonstrated that the fatty acid dehydratase FabZ, the homolog of FabA, undergoes large conformational changes around the active site pocket to enlarge the tunnel for long acyl chain substrate accommodation[5]. The conformational changes also facilitate ACP dissociation after FabZ catalysis in a see-saw manner, and thereby increase its catalytic turnover on long acyl chain substrates. Hence, the conformational changes in FAS-II enzymes upon binding long acyl substrates are a comprehensive process.

In addition to the above interactions, the C2 and C3 hydrogen atoms of the octanoyl chain point towards the $N^{\delta 1}$ atom of H182 and $N^{10}$ atom of FMN with distance of ~2.1 and 2.7 Å (Fig. 4). Previous studies on the catalytic mechanisms of NMO family

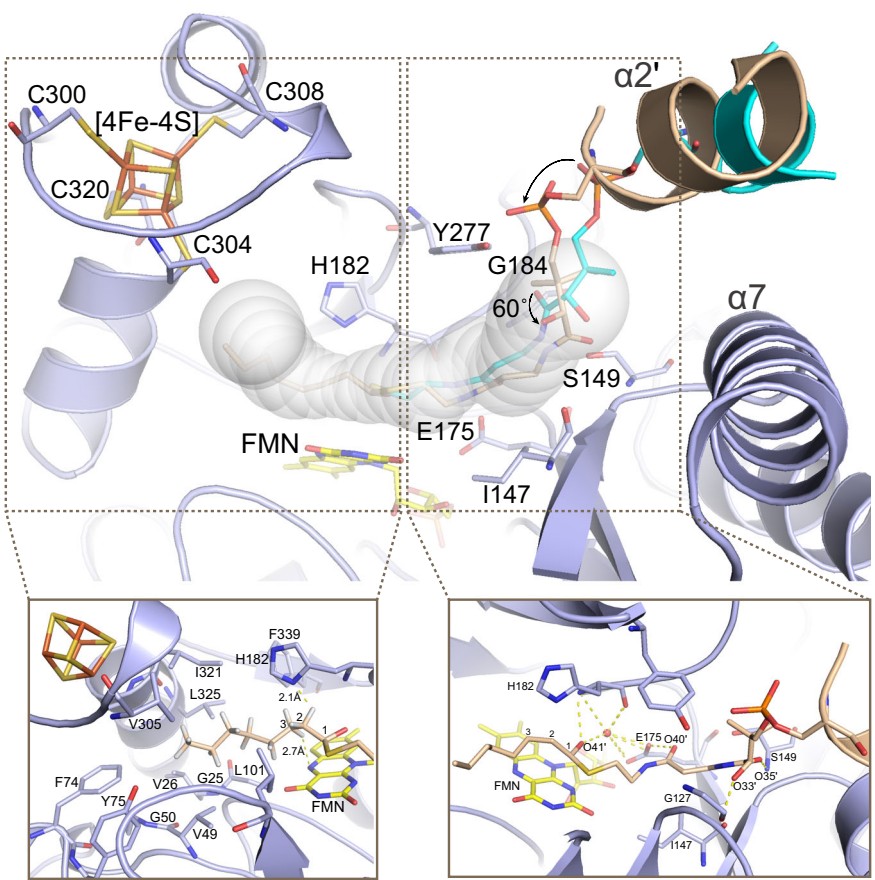

**Fig. 4 The interactions of the octanoyl chain carried by the PPant prosthetic group of ACP inside the FabX tunnel.** Structural superposition of FabX–holo-ACP complex (green/cyan, only ACP was shown) with FabX–octanoyl-ACP complex (light blue/wheat). Residues involved in the interactions were shown in sticks and labeled. The active site tunnel is shown as a transparent white tube. Yellow dashes indicate hydrogen bonds between FabX residues and the acyl chain or PPant moiety.

members such as FabK confirm that such close distances facilitates nucleophilic attack by H182, and allow the subsequent reactions (dehydrogenation in FabX case that generate the *trans*-2-acyl chain)[19–21]. The $N^{\delta1}$ atom of H182 would extract one proton from the substrate, while the other proton and two electrons would be transferred to FMN. However, FMN has three states, the fully oxidized FMN, the semiquinone (FMNH•) intermediate, and the fully reduced FMNH$_2$. However, which state FMN adopts in FabX catalysis remains unclear (see below).

**Mechanism of FabX catalyzed dehydrogenation.** The close distance between FMN and [4Fe–4S] cluster (13.4 Å) indicates that the [4Fe–4S] cluster could be involved in the electron transfer during catalysis[26]. To understand how the [4Fe–4S] cluster contributes to the electron transfer, electron paramagnetic resonance (EPR) spectroscopy was employed. Titration of oxidized FabX using different electron equivalents (with respect to protein) of the reducing reagent sodium dithionite (DT) was carried out to simulate how the electrons are accepted and transferred between FMN and [4Fe–4S] cluster. As shown in Fig. 5a and b, with the addition of increasing electron equivalents of DT, the signals originating from the partially reduced FMN semiquinone radical (FMNH•) and the reduced [4Fe–4S] cluster ([4Fe–4S]$^{1+}$) were observed sequentially. Specifically, when one electron was introduced (one electron equivalents of DT added, red traces), FMN preferentially picked up the electron and became FMNH• radical. At this stage, the FMNH• radical reached it maximum while signal from [4Fe–4S]$^{1+}$ were barely

visible. While an additional electron was introduced (two electron equivalents of DT added, blue traces), signal of [4Fe–4S]$^{1+}$ began to increase notably and the intensity of the FMNH• radical decreased, implying that the second electron was transferred to [4Fe–4S], forming [4Fe–4S]$^{1+}$. At this state, it was speculated that there would be a strong dipolar coupling effect between the two adjacent paramagnetic centers (FMNH• radical and [4Fe–4S]$^{1+}$), which resulted in attenuated signal intensity of the FMNH• radical (compared to the red trace)[29]. Lastly, when the third electron was subsequently introduced (three electron equivalents of DT added, cyan traces), FMN became fully reduced to FMNH$_2$ and was no longer paramagnetic (in other words, EPR-silent). Also, the dipolar coupling effect vanished and the intensity of [4Fe–4S]$^{1+}$ reached its maximum (Fig. 5a, cyan and magenta trace). Since both the FMN and [4Fe–4S] cluster were fully reduced at this state, adding the fourth electron equivalents of DT (magenta traces) brought no distinct spectral change.

These EPR titration results provided a basis for understanding the electron transfer pathway between FMN and the [4Fe–4S] cluster, and suggesting that: (i) one FabX molecule can accept no more than three electrons and became fully reduced; (ii) FMN preferentially will pick up the first electron from the substrate and becomes FMNH• radical; the second electron will be transferred to [4Fe–4S] via the radical. Thus, the [4Fe–4S] cluster has a redox potential lower than that of the FMN/FMNH• pair, but higher than that of the FMNH$_2$/FMNH• pair, resulting in the sequential appearance and maximization of EPR signals from FMNH• radical and [4Fe–4S]$^{1+}$; (iii) the electrons are transferred in the

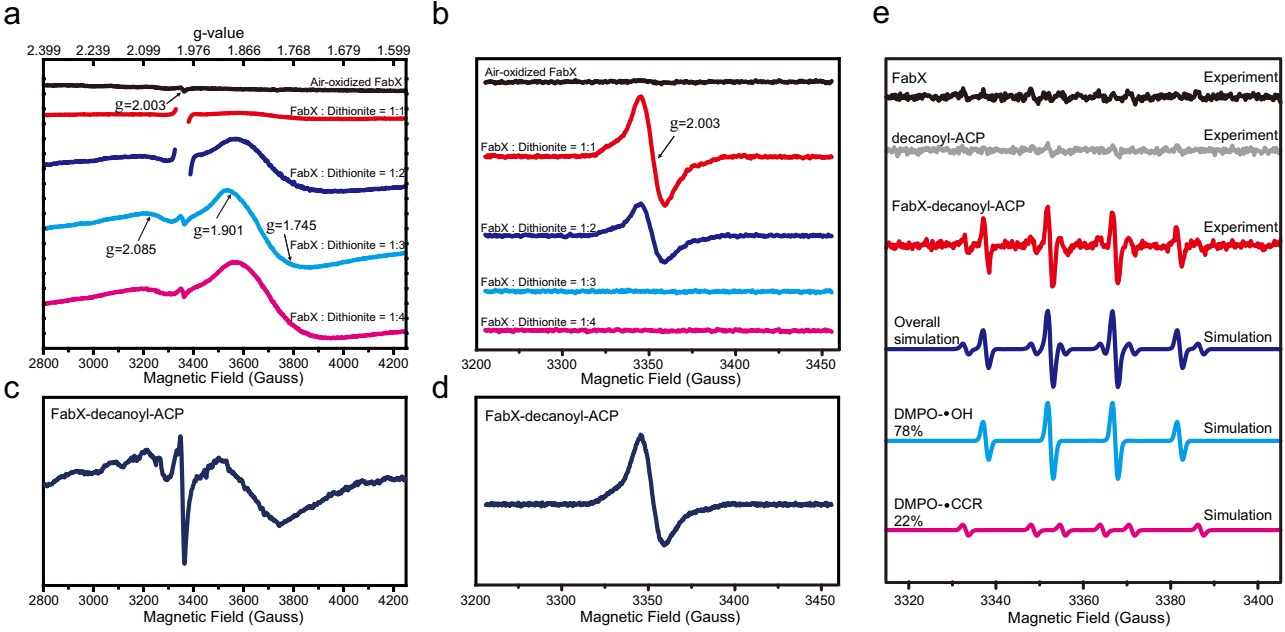

**Fig. 5 EPR spectroscopy.** EPR spectra were collected at (**a**) 10 K and (**b**) 120 K for reductive titration of air-oxidized FabX after addition of different electron equivalents (with respect to FabX protein) of dithionite. FabX purified under aerobic condition showed minor peaks corresponding to FMNH• radical (black trace in **a**). Different electron equivalents (with respect to FabX protein) of dithionite solution (freshly prepared) were titrated to oxidized FabX under anaerobic condition and EPR spectra were collected using same instrumental parameters. Intense radical signals at $g = 2.003$ have been omitted in (**a**) (red and blue trace) to highlight the signal of $[4Fe–4S]^{1+}$. Upon addition of increasing electron equivalents of dithionite, intense signals arose from the FMNH• radical ($g$-value of 2.003) and the $[4Fe–4S]^{1+}$ ($g$-values of 2.085, 1.901, and 1.745). EPR spectra collected at **c** 10 K or **d** 120 K for FabX during catalysis of decanoyl-ACP under aerobic condition. The reactions were quenched by flash-freezing in liquid nitrogen after 20 s of incubation at ambient temperature (298 K) for subsequent EPR measurements. The EPR signals corresponding to FMNH• radical and $[4Fe–4S]^{1+}$ were observed. **e** EPR spin-trapping characterization of FabX-catalyzed dehydrogenation of decanoyl-ACP under aerobic condition. Upper spectra: EPR spectra of DMPO-radical adducts produced from FabX alone (black trace), decanoyl-ACP alone (gray trace), the reaction between FabX and decanoyl-ACP (red trace) are scaled and plotted. The cyan and magenta spectra represent simulations of hydroxyl radical adduct (DMPO–•OH, parameters used for simulation: $g = 2.0062$, $a^N = a^H = 14.84$ G) and carbon-centered radical adduct (DMPO–•CCR, parameters used for simulation: $g = 2.0062$, $a^N = 15.82$ G, $a^H = 22.43$ G), respectively. The blue trace is the sum of the simulations of the two spin adducts with a ratio of DMPO–•OH:DMPO–•CCR = 78%:22%. Experimental spectra were obtained using a microwave of 1 mW, a modulation amplitude of 1 G, modulation frequency of 100 kHz at room temperature (298 K). Source data are provided as a Source Data file.

order:

$$
\begin{aligned}
FMN - [4Fe - 4S]^{2+} &\xrightarrow{+1e^-} FMNH \cdot - [4Fe - 4S]^{2+} \xrightarrow{+1e^-} \\
FMNH \cdot - [4Fe - 4S]^{1+} &\xrightarrow{+1e^-} FMNH_2 - [4Fe - 4S]^{1+} \xrightarrow{-1e^-} \\
FMNH \cdot - [4Fe - 4S]^{1+} &\xrightarrow{-1e^-} FMNH \cdot - [4Fe - 4S]^{2+} \xrightarrow{-1e^-} \\
FMN - [4Fe - 4S]^{2+} &
\end{aligned}
\tag{1}
$$

which is partially similar to the previously reported electron transfer flavoprotein-ubiquinone oxidoreductase (ETF-QO) on the mitochondrial respiratory complex[30–32]. In addition, an oxidative titration of reduced FabX protein using different electron equivalents (with respect to protein) of the oxidative reagent ferricyanide was also carried out (Fig. S27). The oxidative titration gave rise to EPR spectrum in reverse order compared to those from reductive titration using DT, further confirming the electron transfer pathway proposed here. Moreover, signals arising from FMNH• radical and $[4Fe–4S]^{1+}$ were also observed during dehydrogenation of decanoyl-ACP catalyzed by FabX (Fig. 5c and d), demonstrating successful electron transfer from decanoyl-ACP to FabX that is consistent with the observations from the titration assay. The [4Fe–4S] cluster of FabX might function as a temporary electron depot in a way similar to the N1a cluster of the mitochondrial respiratory complex I (NADH-quinone oxidoreductase)[33,34]. Hence, the function of FMN and

[4Fe–4S] cluster in FabX catalysis is more like a combination of ETF-QO and complex I.

**FabX dehydrogenation produces reactive oxygen species.** Previously we reported that $H_2O_2$ was produced during FabX catalysis[18]. However, reaction with assay components inhibited quantitative detection of $H_2O_2$ and superoxide was not assayed. Hence, we have turned to EPR spin-trapping method to measure ROS production. The FabX-catalyzed dehydrogenation of decanoyl-ACP was studied by EPR spin-trapping in the presence of the spin trap 5,5-dimethyl-pyrroline N-oxide (DMPO) to allow formation of more stable paramagnetic adducts. This approach indicated that high levels of free radicals were formed in the FabX dehydrogenation reaction. Experimental and simulated EPR spectra of the radical adducts are presented in Fig. 5e. Substantial formation of hydroxyl radical adduct (DMPO–•OH) was observed in the dehydrogenation reaction (Fig. 5e, blue trace), along with a small amount of a carbon centered radical adduct (DMPO–•CCR) which might originate from hydrolysis of DMPO[35,36]. The prominent presence of hydroxyl radical adducts (DMPO–•OH) can result from either direct trapping of the hydroxyl radical (•OH) or by rapid breakdown of DMPO–•OOH, the adduct formed by superoxide[37,38]. While it is difficult to distinguish whether the DMPO–•OH adduct observed here corresponds to trapping of superoxide or hydroxyl radical, in many other flavin enzymes, the primary ROS produced is superoxide.

Sometimes the superoxide diffuses away but often another electron transfer occurs before escape of the first superoxide and generates a second superoxide molecule that combines with the first to form $H_2O_2$. Hence, flavin enzymes generally produce a mixture of the two ROS species[39]. Note that the diamagnetic $H_2O_2$ is EPR silent.

We also performed a sensitive fluorescent hydrogen peroxide assay to directly measure hydrogen peroxide produced by *H. pylori* strains. As shown in Fig. S28, the genetic replacement of FabX by FabA in *H. pylori* resulted into 8.9 and 2.1-fold decrease in $H_2O_2$ level released to media, when detected in cultures of $OD_{600}$ of 0.3 and 1.0, respectively. Given that FabA performs the dehydrase and isomerase reactions without involvement of ROS production, the superoxide produced by FabX catalysis is a major resource of peroxide excreted by *H. pylori*. This may play a key role in pathogenesis by corrosion of gastric mucosa.

## Discussion

In our proposed mechanism, FabX recognizes and binds the α2 and α3 helixes of the substrate ACP moiety via its α7 helix. The ACP moiety rotates along its α2 helix and inserts the substrate decanoyl chain and thioester-proximal portion of the attached PPant prosthetic group into the active site tunnel. The $N^{\delta 1}$ atom of H182 performs nucleophilic attack on a hydrogen atom attached to C2 of the decanoyl chain and extracts one proton from the substrate. The other proton and two electrons are transferred to FMN. One pair of proton/electrons is held by FMN, forming the reduced FMNH• intermediate, while the other electron is transferred to and stored on the [4Fe–4S] cluster located ~13.4 Å away (Fig. 6a–c)[26,34]. According to the previous report which analyzed Protein Data Bank (PDB) enzymes whose function involve electron transfer having known atomic structures, electrons can travel up to 14 Å between redox centers through the protein medium[26]. In FabX, the distance between FMN and the [4Fe–4S] cluster is 13.4 Å, and thereby the electrons could more likely transfer directly between the FMN and the [4Fe–4S] cluster. Simultaneously, the *trans*-2-acyl-ACP intermediate isomerizes to *cis*-3-decenoyl-ACP (Fig. 6d). After isomerization, the *cis*-3-decenoyl-ACP leaves FabX, while oxygen from the bulk solution enters the active tunnel and is locked in the oxyanion hole, oxidizing the FMNH• and generating a ROS molecule (Fig. 6e–g). Subsequently, the superoxide molecule diffuses from the oxyanion hole, and a second oxygen is captured by the oxyanion hole. Meanwhile, the electron stored in the [4Fe–4S] cluster transfers back to FMN, reducing it to FMNH• again (Fig. 6g, h). Afterward, it is again oxidized by oxygen, generating a second superoxide molecule. The superoxide molecules are subsequently released from the oxyanion hole to bulk solution either as such or more probably will combine to form $H_2O_2$. Meanwhile, the protons that were held on FMN and His182 are re-equilibrated to bulk solution (Fig. 6i). The two cofactors return to original oxidized status, and FabX is ready for the next round of decanoyl-ACP binding and catalysis.

Flavin semiquinones are more readily oxidized than the fully reduced flavin[40,41] and thus the primary role of the [4Fe–4S] cluster is to boost the catalytic efficiency of FabX. In the case of FabX, no final electron acceptor other than oxygen has been found. Indeed, the oxygen concentration giving optimal FabX activity matches that giving optimal growth of the microaerophilic *H. pylori*[18]. However, since oxygen molecules can only receive one electron, the electron accepting oxygen molecule must reside in the active site for successful electron transfer as seen in the LOX reaction (Fig. 2e)[22]. The acyl chain of the acyl-ACP

substrate of FabX appears to block oxygen binding which may explain the requirement for electron storage on the [4Fe–4S] cluster.

During dehydrogenation catalysis by FabX, oxygen is required as the final electron acceptor. However, the maintenance of the reduced status of the [4Fe–4S] cluster is required for its activity during electron transfer. High concentrations of oxygen or the superoxide/ROS products inside the cell is detrimental for its integrity as the cluster is located very close to the protein surface. Hence, on one hand, FabX homologs mostly exist in microaerophilic bacteria to avoid a high concentration of oxygen in the environment. On the other hand, *H. pylori* utilizes the low concentration of oxygen to generate superoxide/ROS and excretes it to extracellular environment to protect FabX from over-oxidation and achieve its pathogenic function in the corrosion of gastric mucosa.

Following dehydrogenation of the substrate, there is strong evidence that isomerization of the *trans*-2-species to the *cis*-3-species proceeds spontaneously as dictated by the shape of the enzyme tunnel, and oxygen is not required for the isomerization[8,42]. The well-studied *E. coli* FabA functions as a dehydratase/isomerase, while its highly conserved homolog *E. coli* FabZ is only a dehydratase[42]. Comparison of the structures of FabA and FabZ indicates that tunnel shape is the key to spontaneous isomerization[8,42]. The FabX active site tunnel has a mildly kinked "L" shape similar to that of FabA whereas the active site tunnel of FabZ has a tortuous "U" shape (Fig. S29). Hence, the kinked "L" shape of the FabX and FabA tunnels provide the proper angle for the C3–C4 dihedral shift required for the spontaneous *trans*-2 to *cis*-3 isomerization of the decenoyl-ACP acyl chain.

Ongoing increase in antibiotic resistance has led the World Health Organization to list *H. pylori* as a high priority pathogen for which advanced antibiotics are urgently needed. The bacterial type II fatty acid synthesis pathway is regarded as an attractive target because it is distinct from the type I fatty acid synthesis pathway of mammals. However, the well-studied enzymes involved in SFA synthesis are largely conserved in bacteria and thus inhibition of these enzymes could theoretically disrupt the healthy gut microbiome. In contrast to SFA synthesis, UFA synthesis is not only essential for most bacteria but proceeds by diverse mechanisms that should allow species-specific inhibitors. In 2016, we discovered the atypical dehydrogenation/isomerase bifunctional enzyme FabX[18]. We have now shown that FabX is an essential enzyme in *H. pylori* UFA synthesis as well as in ROS production that may be involved in pathogenesis, and thus is a valid target for antibiotic development. Granted that other bacteria encode FabX homologs, most of these are environmental isolates rather than human microbiome bacteria.

*H. pylori* has been reported to produce excessive ROS that may corrode the human gastric mucosa and facilitate invasion by the bacterium. Several mechanistic hypotheses have been put forth[43–45]. We have uncovered FabX as a major source of ROS in *H. pylori* that may play a role in infection. Note that *H. pylori* has the means to prevent damage from its own ROS as well as that from the strong host inflammatory response. *H. pylori* has a well characterized superoxide mutase and a robust catalase as well as other enzymes that catalyze reduction of hydrogen peroxide, peroxynitrite, and a wide range of organic hydroperoxides (ROOH) to their corresponding alcohols[46].

Bioinformatic studies suggest that FabX homologs (identity higher than 50%) are encoded in 1650 species belonging to 31 genera in 7 phyla. Most of these bacteria live under anaerobic conditions and rarely associate with human healthy gut flora (Fig. S30). Sequence alignment among FabX and the homologs delegated from the phyla suggests that they share

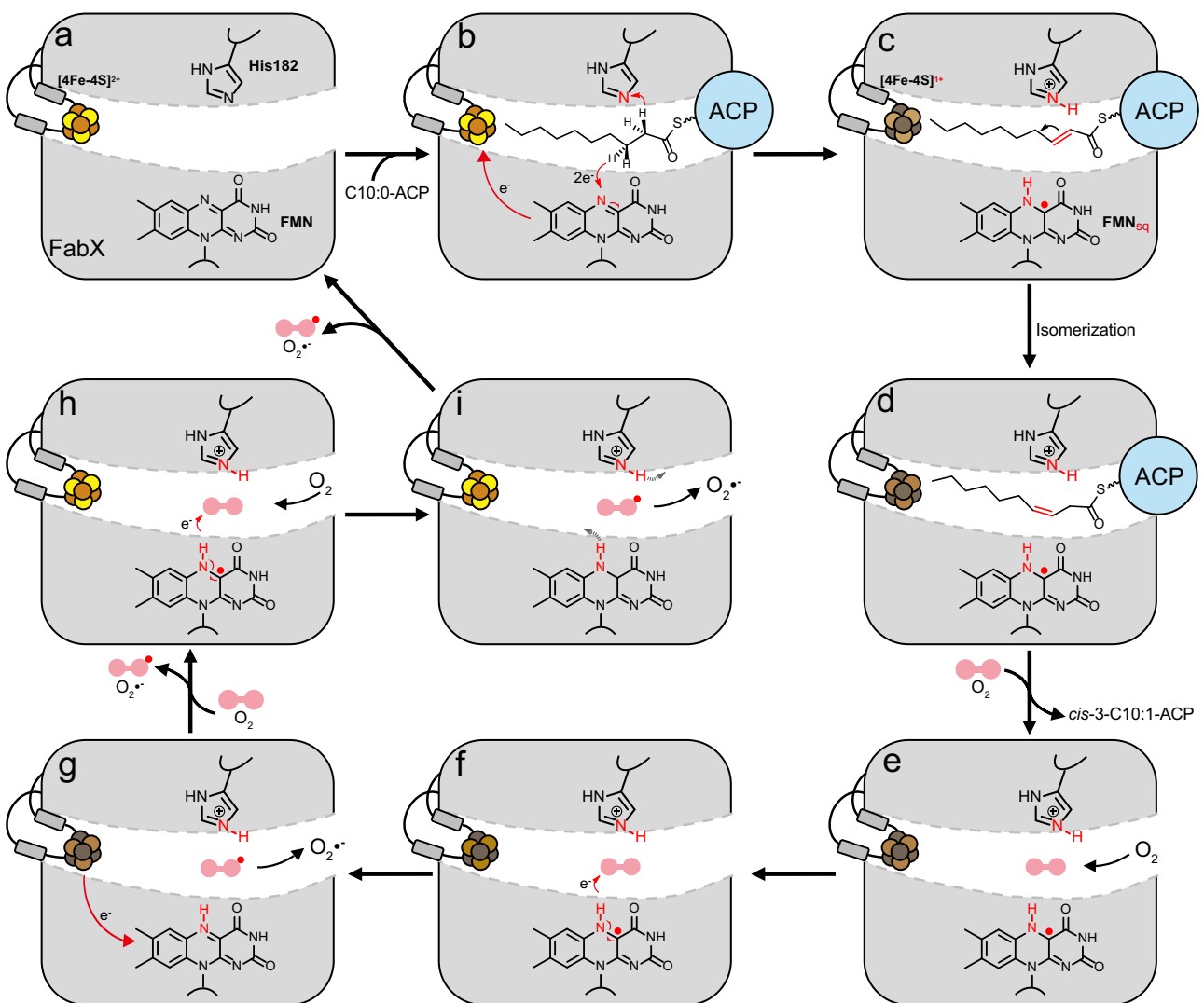

**Fig. 6 The proposed catalytic mechanism of FabX.** FabX and ACP were depicted in gray and light blue, respectively. The [4Fe–4S] cluster and $O_2$ molecule are depicted in orange tube and red spheres, respectively. The chemical structures of FMN, H182, and decanoyl chain are shown, and the key hydrogen and nitrogen atoms involved in the catalysis are colored in red. The red and black arrows indicate the proton or electron transfer directions, respectively. **a** The active site tunnel of FabX. The [4Fe–4S] cluster blocks the exit of the tunnel, while FMN is located at the middle of the tunnel. **b** ACP binds to FabX, and inserts the decanoyl chain into the hydrophobic active tunnel. The $N^{\delta1}$ of H182 and $N^{10}$ of FMN atoms regioselective nucleophilic attack the hydrogen atoms located at the 2 and 3-positions of the acyl chain, respectively. Meanwhile, an electron–protons pair is captured by FMN to generate the FMNH•, while the other electron is transferred to the [4Fe–4S] cluster for temporary storage. The *trans*-2-decenoyl-ACP intermediate is generated. **c** and **d** The *trans*-2-decenoyl intermediate spontaneously isomerizes to *cis*-3-decenoyl-ACP, and subsequently releases from FabX. **e** The $O_2$ molecule enters the tunnel (black arrows), and is locked in the oxyanion hole between H182 and FMN. **f** The $O_2$ molecule oxidizes the FMNH•, generating a superoxide molecule. **g** The superoxide molecule is released from the oxyanion hole, and the electron stored on the [4Fe–4S]$^{1+}$ is transferred back to FMN where it again reduces the flavin to the FMNH•. **h** A second $O_2$ molecule enters the tunnel (black arrows) and is locked in the oxyanion hole. It oxidizes the FMNH•, generating a second superoxide molecule. **i** The superoxide molecule releases from the oxyanion hole either as such or combining to $H_2O_2$. Meanwhile, the protons that were held on FMN and His182 are re-equilibrated to bulk solution (gray dashed arrows), and the two cofactors return to original oxidized status. FabX is ready for the next round of decanoyl-ACP binding and catalysis.

highly conserved catalytic motifs (Fig. S25). This information suggests that FabX is a potentially specific anti-*H. pylori* drug target.

In this study, we have elucidated the mechanisms of *H. pylori* FabX in recognizing, regulating and catalyzing synthesis of *cis*-3-decenoyl-ACP, the essential UFA biosynthesis intermediate, which deepens our understanding of UFA biosynthesis in FAS-II systems. Our work may facilitate drug discovery against FabX, and could provide a "precision medicine" strategy against *H. pylori* and the attendant gastric cancers.

## Methods

**Bacterial strains and culture conditions**. *H. pylori* strains used in this study, summarized in Table S3 were routinely cultured either in brain heart infusion broth (BHI, Becton Dickinson, Sparks, MD, USA) medium containing 10% fetal calf serum (FCS) or on Columbia blood agar (Oxoid, Basingstoke, UK) plates containing 5% FCS, supplemented with Dent selective supplement (Oxoid). All plates or media were incubated at 37 °C for 48–72 h under microaerophilic conditions (10% $CO_2$, 85% $N_2$, and 5% $O_2$ and 90% relative humidity) using a double-gas $CO_2$ incubator (Binder, model CB160, Germany). The standard strain G27 was kindly provided by Dr. Nina R. Salama, Fred Hutchinson Cancer Research Center, USA. *S. pneumonia* ATCC 49619 was cultured as *H. pylori* with the same broth medium and growth condition for hydrogen peroxide measurement.

**Protein subcloning, expression and purification.** The expression and purification of FabX, apo-ACP and holo-ACP from *H. pylori*, FabI from *E. coli*, and acyl-ACP synthetase (AasS) from *Vibrio harveyi* were described previously[5,18,47,48]. Briefly, FabX, ACP, FabI and AasS genes were subcloned to pQE2, pET32a, pET28b and pET16b vectors respectively. QuickChange Site-Directed Mutagenesis (Stratagene) strategy was used to perform FabX mutagenesis with pBHK614 as the PCR template[18]. The primers used in PCR and mutagenesis are listed in Table S3. The constructed plasmids were transformed into *E. coli* DH5α by CaCl₂ treatment. The mutations were verified by DNA sequencing. The plasmids were transformed and expressed in BL21(DE3) Rosetta cells, and were purified by using Ni-NTA affinity column. Subsequently, FabX and AasS was purified by using HiTrap-SP or HiTrap QFF anion exchange columns, while the N-terminal thioredoxin-His tag of ACP and N-terminal SUMO-His tag of AasS were removed through cleavage with recombinant TEV or Ulp1 protease, respectively. FabX, apo-ACP, holo-ACP, FabI and AasS proteins were further purified by using GE HiLoad 16/600 Superdex75 prep grade gel-filtration columns with running buffer containing 20 mM HEPES (pH 8.0), 150 mM NaCl and 1 mM DTT. For the FabX–ACP complex, the purified FabX and holo-ACP proteins were mixed in a molar ratio 1:1.2 and loaded to HiLoad 16/600 Superdex75 prep grade column for further purification. The peaks from the gel-filtration column were further pooled and concentrated, and the final concentration of the proteins was quantified by the Bradford reagent purchased from Bio-Rad.

**Acyl-ACP synthesis and purification.** Acyl-ACP (hexanoyl-, octanoyl-, decanoyl-, and dodecanoyl-ACP) was synthesized by using AasS enzyme mediated in vitro biochemical reactions[4,49]. The acylation reaction mixture contains 100 mM Tris–HCl (pH7.8), 10 mM MgCl₂, 10 mM ATP, 1 mM DTT, 750 μM fatty acid (sodium hexanoate, sodium octanoate, sodium decanoate or sodium dodecanoate), 75 μM holo-ACP and 5 μM AasS in a total volume of 1.8 ml. Then, the reaction mixtures were placed in a 37 °C incubator for 5 h[14,50]. Subsequently, the mixture was diluted five times with a buffer containing 20 mM MES, pH 6.0, 100 mM LiCl, and then loaded to HiTrap QFF anion exchange column for further purification. The purified Acyl-ACP was pooled, and dialyzed to buffer containing 20 mM Tris, pH 7.5. The proteins were further concentrated and the final protein concentrations were quantified. The acylation modifications were confirmed by using LC–MS/MS (Fig. S3).

**FabX–FabI coupled enzymatic and kinetics characterization.** The enzymatic activity of FabX and FabX mutants were determined by coupling *E. coli* FabI to FabX as described previously[18]. The reaction mixtures contains 50 mM sodium phosphate (pH 7.2), 50 mM NaCl, 0.25 μM FabX (1.25 μM for decanoyl-CoA reaction and C304A at 2 μM), 1 μM FabI (5 μM for decanoyl-CoA reaction), 125 μM NADH and 4 μM decanoyl-ACP in a total volume of 25 μl. In the selective experiment of carbon chain length, 1 μM FabX and 4 μM acyl-ACP (hexanoyl-, octanoyl-, decanoyl-, or dodecanoyl-ACP) were used to determine the substrate preferences under 25 °C. The reaction was measured either by monitoring the decrease in absorbance at 340 nm (NADH) using a UV Synergymx (Biotek) at 25–37 °C, or by conformationally sensitive gel electrophoresis. For monitoring the absorbance changes, the kinetics was determined by using decanoyl-ACP (0.25, 0.5, 1, 2, 4, 5, 8, or 10 μM) or substrate analog decanoyl-CoA (30, 60, 100, 150, 200, or 250 μM). All enzyme kinetics experiments were conducted in triplicate, and the obtained data have been fitted using nonlinear regression analysis of Graphpad Prism 8 software (Table S1). For monitoring gel electrophoresis, the reaction was initiated by adding decanoyl-ACP and incubated at 37 °C for 20 min. Then the reaction was sampled, immediately quenched by addition of an equal volume of 10 M urea, and stored on dry ice until analysis. The reaction samples were mixed with gel loading buffer and analyzed by conformationally sensitive gel electrophoresis on 18% polyacrylamide gels containing a urea concentration optimized for the separation[51].

**Isothermal titration calorimetry (ITC) assay.** The binding affinities of holo-ACP to FabX or FabX R164A were determined by MicroCal PEAQ-ITC (Malvern) at 10 °C. The purified FabX (or FabX R164A) and holo-ACP protein were dialyzed into buffer containing 20 mM HEPES (pH 8.0), 150 mM NaCl, 1 mM TCEP, and concentrated. Subsequently, FabX (or FabX R164A) was loaded into the cell, while holo-ACP was loaded into the syringe at the molecular ratio of 1:10 (FabX:holo-ACP). A default PEAQ-ITC method was used, with an initial injection of 0.4 μl holo-ACP followed by 19 injections of 2 μl at 150 s intervals. The analysis of raw data and the determination of $K_d$ value were performed by MicroCal PEAQ-ITC Analysis Software (Malvern).

**Protein thermal shift assay (PTSA).** PTSA was performed by using LightCycler 480 II (Roche). FabX or FabX mutants were diluted with a storage buffer and mixed with 5X SYPRO Orange dye (Sigma-Aldrich) to a final concentration of 4 μM. After incubation, 20 μl of samples were added to a 96-well plate and centrifugation. The thermal shift program was set to 25–95 °C for 30 min. Subsequently, the melt curve was analyzed by LightCycler thermal shift analysis software (Roche) and the $T_m$ value was fitted.

**FMN absorption screening.** Cofactor FMN absorption screening was conducted by BioMate 3S UV–visible spectrophotometer (Thermo Fisher Scientific) at 25 °C. FabX or FabX mutants were diluted to 10 μM. Subsequently, the characteristic absorption peak of FMN (444 nm) was determined by ultraviolet wavelength scanning.

**LC–MS/MS analysis of FMN content.** The FabX cofactor FMN for its mutants was obtained by heating 100 μM protein at 95 °C for 10 min. The precipitate was removed by centrifugation at 13,000×*g* for 15 min. The content of FMN was subsequently determined by liquid chromatography–mass spectroscopy according to our previously study[18]. The LC–MS/MS analysis was performed by Nexera UHPLC LC-30A system (Shimadzu) coupled to SCIEX API 4000 QQQ mass spectrometera (SCIEX). 5 μl samples were injected to Agilent ZORBAX Stablebond-Aq (RRHT) C18 (1.8 μm, 2.1*100 mm) column with mobile phase A (10 mM ammonium formate) and mobile phase B (100% acetonitrile) at 0.2 ml/min flow rate. Using the MRM mode with negative-ion detection, the FMN precursor ion was 455.0, and the product ion was 97.0. Declustering potential (DP) voltage and collsion energy (CE) voltage were −80, −40eV; The ion spray voltage was conducted at −4.5 kV.

**EPR measurement.** Low-temperature EPR spectra were acquired on a Bruker X-band EMX plus 10/12 spectrometer equipped with an Oxford Instruments ESR 910 continuous helium-flow cryostats. A cylindrical resonator (ER4119hs TE011) was used for EPR data collection. Samples were prepared as in "air-oxidized" state (FabX as purified), "reduced" state (either reduced by sodium DT in an anaerobic chamber (Coy Laboratory Products) or by decanoyl-ACP). 10% glycerol was added to all samples as cryoprotective agent and each sample was placed into quartz EPR tubes (Wilmad, 707-SQ-250 M, 3 mm i.d., 4 mm o.d.). For each sample, multiple scans were accumulated to obtain a good S/N ratio. Experimental conditions: temperature, 10 K for EPR characterization of [4Fe–4S] cluster and 120 K for detection of radical signal; microwave power, 2 mW at 10 K and 10 μW at 120 K (to avoid power saturation of the semiquinone radical); modulation amplitude, 5 G at 10 K and 2 G at 120 K; modulation frequency, 100 kHz; resonance frequency, ~9.398 GHz.

Reductive titration of oxidized FabX and oxidative titration of reduced FabX were both carried out under anaerobic condition. Specially, reduction of the protein was accomplished by progressive addition of different electron equivalents (with respect to protein) of anaerobically prepared sodium DT to oxidized FabX and EPR spectrum was subsequently collected. For the oxidative titration, FabX was first anaerobically reduced with freshly prepared sodium DT and excess DT was removed using desalting columns. The reduced protein was then progressively reoxidized by adding increasing molar equivalents (with respect to protein) of ferricyanide (K₃Fe(CN)₆), while the signal of reduced [4Fe–4S]¹⁺ and FMNH• radical were monitored using EPR spectroscopy.

For EPR spin-trapping experiments, 5,5-dimethyl-1-pyrroline-N-oxide (DMPO, Dojindo Laboratories, Kumamoto, Japan, 200 mM final concentration) was used as spin-trapping reagent. Samples were loaded into a borosilicate glass capillary (Kimble Chase Life Science and Research Products, LLC.) for the EPR measurement. EPR spectrum of the samples were recorded at room temperature using a Bruker X-band EMX A300 spectrometer equipped with a high-sensitivity ER4119HS resonator. Experimental parameters were 1 mW incident microwave power, 100 kHz field modulation, and 0.5 G modulation amplitude. For each sample, 20 scans were accumulated to obtain a good S/N. EPR simulation was performed by the program EasySpin ver. 5.2.28 operated in Matlab[52].

**Size-exclusion chromatography (SEC) coupled with multi-angle light scattering (MALS) measurements.** The absolute molar masses of FabX, holo-ACP and FabX–holo-ACP complex were determined by SEC MALS. In order to ensure the stability, 200 μl of 2 mg/ml BSA was firstly injected into Wyatt Technology WTC-030S5 column connected to HPLC system (Agilent Technologies) as a pre-experiment with the mobile phase containing 20 mM HEPES (pH 8.0), 50 mM NaCl and 10 mM TCEP. Next, 10 mg/ml FabX, holo-ACP or FabX–holo-ACP complex was injected into the system, respectively, and separated by WTC-030S5 column. The light scattering analysis software ASTRA (Wyatt Technology) was used to evaluate and analyze the corresponding molecular weight of the fraction peaks.

**FabX, FabX–holo-ACP and FabX–octanoyl-ACP complex crystallization.** Purified FabX (1 μl of 30 mg/ml) or FabX–holo-ACP (1 μl of 25 mg/ml) was mixed with an equal volume of reservoir solution and equilibrated against 100 μl of the reservoir solution at 292 K. Brown plate-shaped crystals were observed about 6 days in the reservoir solution containing 0.1 M Tris–HCl (pH 7.0) and 25% w/v PEG3350 (for FabX) or 0.1 M HEPES (pH 7.0) and 20% w/v PEG 8000 (for FabX–holo-ACP). For FabX–octanoyl-ACP complex, 35 mg/ml purified FabX and octanoyl-ACP proteins were mixed in molar ratio 1:1.2 and incubated for 3 h under anaerobic condition. The complex was further mixed with an equal volume of reservoir solution and equilibration against 100 μl of the reservoir solution at 298 K. Brown-plate shaped crystals were observed around 4 days in the reservoir solution containing 0.1 M MES pH 7.0 and 20% w/v PEG4000. The crystals were

soaked in protectant containing mother liquor with 30% glycerol (v/v) and flash frozen for X-ray diffraction.

**Data collection and structural determination.** The crystal diffraction data of FabX, FabX–holo-ACP, FabX–octanoyl-ACP were collected at BL19U1 beamline of National Facility for Protein Science in Shanghai (NFPS) at Shanghai Synchrotron Radiation Facility (SSRF), and processed with HKL3000 or autoPROC[53–55] (Table S2). Molecular replacement was used to solved the structures with the TIM-barrel domain structure of *P. gingivalis* FabK (pdb code: 4IQL) and *H. pylori* holo-ACP structure (pdb code: 5H9H) as search models, respectively. Subsequently, the data file was refined by program Phenix and the model building was conducted via computer graphics program Coot[56,57]. The protein structure viewer was performed by PyMol software. The atomic coordinates and structure factors for the reported crystal structures were deposited under accession codes 7E1Q, 7E1R, and 7E1S.

**DNA manipulation and construction of strains.** Plasmid and chromosomal DNAs were extracted with the QIAprep spin miniprep kit (Qiagen). Oligonucleotide primers (Table S3) and DNA fragments were synthesized and the cloned genes were verified by sequencing performed by GeneScript Co. (China). Strains BHKS487 and BHKS488 were constructed first by inserting the wild type *fabX* and the mutant *fabX R164A* (containing the *ureI* promoter) into an improved pIR203C04 complementation system[58], respectively. The two regions were synthesized and ligated to pBHKP252 carrying a sandwich fusion in which the *catGC* cassette was flanked by the upstream and downstream regions of the *hp0203–204* intergenic region[59], to generate pBHKP405 and pBHKP406, respectively. The two plasmids were then introduced into strain G27 by natural transformation via allelic exchange, and strains BHKS485 and BHKS486 were isolated on Columbia blood agar plates supplemented with chloramphenicol (10 μg/ml). The *fabX* knockout in the chromosome of BHKS485 and BHKS486 was achieved via another allelic exchange with the introduction of pBHKP408, carrying a sandwich fusion in which the *aphA3* cassette was flanked by the upstream and downstream regions of the *fabX* gene. Finally, strains BHKS487 (G27 IR203::*fabX* Δ*fabX*) and BHKS488 (G27 IR203::*fabX R164A* Δ*fabX*) were isolated on Columbia blood agar plates supplemented with kanamycin (25 μg/ml) and chloramphenicol (10 μg/ml). Insertion and deletion of *fabX* or *fabX R164A* in the genome were confirmed by PCR using appropriate primers (Table S3), followed by sequencing of the PCR products.

Strains BHKS551 (NSH57 IR203::*fabA* Δ*fabX*) and BHKS568 (NSH57 IR203::*fabX* Δ*fabX*) were constructed by the same method described above. First the DNA fragments containing *E. coli fabA* optimized for *H. pylori* codons and *fabX* with the *ureI* promoter[60] were synthesized and ligated to pBHKP252 (Table S3), to generate pBHKP421 and pBHKP432, respectively. The plasmids were then introduced into strain NSH57 by natural transformation, and BHKS550 (NSH57 IR203::*fabA*) and BHKS567 (NSH57 IR203::*fabX*) were isolated on Columbia blood agar plates supplemented with chloramphenicol (10 μg/ml). The *fabX* knockout in the chromosomes of the two strains was achieved via another allelic exchange with the introduction of pBHKP408. The strains BHKS551 (NSH57 IR203::*fabA* Δ*fabX*) and BHKS568 (NSH57 IR203::*fabX* Δ*fabX*) were isolated on Columbia blood agar plates supplemented with kanamycin (25 μg/ml) and chloramphenicol (10 μg/ml). The successful construction was confirmed by PCR using appropriate primers (Table S3), followed by sequencing of the PCR products.

**Hydrogen peroxide measurement.** The $H_2O_2$ amounts produced by *H. pylori* and *S. pneumonia* strains were determined by the Amplex® Red Hydrogen Peroxide/Peroxidase Assay Kit from Invitrogen according to the manufacturers' instructions with a $H_2O_2$ standard curve. The strains were cultured in BHI broth media containing 10% FCS to an $OD_{600}$ of 0.3 and 1.0, respectively. The culture supernatants were collected after bacterial cells were removed by centrifugation at 5000×g for 15 min, and directly subjected to $H_2O_2$ quantification in 96-well microplate format with the Amplex Red reagent (10-acetyl-3,7-dihydroxyphenoxasine). The fluorescence measurements were conducted at an emission wavelength of 590 nm and an excitation wavelength of 530 nm, using a Synergy HTX multimode microplate reader (BioTek Instruments, Winooski, VT, USA). Each of the supernatant samples, and the standard curve, were tested in triplicate.

**Reporting summary.** Further information on research design is available in the Nature Research Reporting Summary linked to this article.

## Data availability
The data are available within the paper and its Supplementary Information files. The diffraction data generated in this study have been deposited in the PDB database under accession code 7E1Q (FabX), 7E1R (FabX–holo-ACP) and 7E1S (FabX–octanoyl-ACP). Source data are provided with this paper.

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

# ARTICLE

30. Thorpe, C. & Kim, J. J. Structure and mechanism of action of the acyl-CoA dehydrogenases. *FASEB J.* **9**, 718–725 (1995).

31. Ruzicka, F. J. & Beinert, H. (1978). A new iron sulfur flavoprotein of the respiratory chain. *J. Biol. Chem.* **252**, 8440–8445 (1978).

32. Zhang, J., Frerman, F. & Kim, J. Structure of electron transfer flavoprotein-ubiquinone oxidoreductase and electron transfer to the mitochondrial ubiquinone pool. *Proc. Natl Acad. Sci. USA* **103**, 16212–16217 (2006).

33. Berrisford, J. M., Baradaran, R. & Sazanov, L. A. Structure of bacterial respiratory complex I. *Biochim. Biophys. Acta* **1857**, 892–901 (2016).

34. Ohnishi, T., Ohnishi, S. T. & Salerno, J. C. Five decades of research on mitochondrial NADH–quinone oxidoreductase (complex I). *Biol. Chem.* **399**, 1249–1264 (2018).

35. Bosnjakovic, A. & Schlick, S. Spin trapping by 5,5-dimethylpyrroline-N-oxide in Fenton media in the presence of Nafion perfluorinated membranes: limitations and potential. *J. Phys. Chem. B* **110**, 10720–10728 (2006).

36. Kalyanaraman, B., Janzen, E. G. & Mason, R. P. Spin trapping of the azidyl radical in azide/catalase/$H_2O_2$ and various azide/peroxidase/$H_2O_2$ peroxidizing systems. *J. Biol. Chem.* **260**, 4003–4006 (1985).

37. Zweier, J. L., Broderick, R., Kuppusamy, P., Thompson-Gorman, S. & Lutty, G. A. Determination of the mechanism of free radical generation in human aortic endothelial cells exposed to anoxia and reoxygenation. *J. Biol. Chem.* **269**, 24156–24162 (1994).

38. Irani, K. et al. Mitogenic signaling mediated by oxidants in Ras-transformed fibroblasts. *Science* **275**, 1649–1652 (1997).

39. Imlay, J. A. The molecular mechanisms and physiological consequences of oxidative stress: lessons from a model bacterium. *Nat. Rev. Microbiol.* **11**, 443–454 (2013).

40. Massey, V. Activation of molecular oxygen by flavins and flavoproteins. *J. Biol. Chem.* **269**, 22459–22462 (1994).

41. Mattevi, A. To be or not to be an oxidase: challenging the oxygen reactivity of flavoenzymes. *Trends Biochem. Sci.* **31**, 276–283 (2006).

42. Kimber, M. S. et al. The structure of (3R)-hydroxyacyl-acyl carrier protein dehydratase (FabZ) from *Pseudomonas aeruginosa*. *J. Biol. Chem.* **279**, 52593–52602 (2004).

43. Nagata, K. et al. *Helicobacter pylori* generates superoxide radicals and modulates nitric oxide metabolism. *J. Biol. Chem.* **273**, 14071–14073 (1998).

44. Handa, O., Naito, Y. & Yoshikawa, T. *Helicobacter pylori*: a ROS-inducing bacterial species in the stomach. *Inflamm. Res.* **59**, 997–1003 (2010).

45. Nakamura, A. et al. Oxidative cellular damage associated with transformation of *Helicobacter pylori* from a bacillary to a coccoid form. *Free Radic. Biol. Med.* **28**, 1611–1618 (2000).

46. Wang, G., Alamuri, P. & Maier, R. J. The diverse antioxidant systems of *Helicobacter pylori*. *Mol. Microbiol.* **61**, 847–860 (2006).

47. Sivaraman, S., Zwahlen, J., Bell, A. F., Hedstrom, L. & Tonge, P. J. Structure–activity studies of the inhibition of FabI, the enoyl reductase from *Escherichia coli*, by Triclosan: kinetic analysis of mutant FabIs. *Biochemistry* **42**, 4406–4413 (2003).

48. Jiang, Y., Chan, C. H. & Cronan, J. E. The soluble acyl-acyl carrier protein synthetase of *Vibrio harveyi* B392 is a member of the medium chain acyl-CoA synthetase family. *Biochemistry* **45**, 10008–10019 (2006).

49. Jiang, Y., Morgan-Kiss, R. M., Campbell, J. W., Chan, C. H. & Cronan, J. E. Expression of *Vibrio harveyi* acyl-ACP synthetase allows efficient entry of exogenous fatty acids into the *Escherichia coli* fatty acid and lipid a synthetic pathways. *Biochemistry* **49**, 718–722 (2010).

50. Agarwal, V., Lin, S., Lukk, T., Nair, S. K. & Cronan, J. E. Structure of the enzyme–acyl carrier protein (ACP) substrate gatekeeper complex required for biotin synthesis. *Proc. Natl Acad. Sci. USA* **109**, 17406–17411 (2012).

51. Post-Beittenmiller, D., Jaworski, J. G. & Ohlrogge, J. B. In vivo pools of free and acylated acyl carrier proteins in spinach. Evidence for sites of regulation of fatty acid biosynthesis. *J. Biol. Chem.* **266**, 1858–1865 (1991).

52. Stoll, S. & Schweiger, A. EasySpin, a comprehensive software package for spectral simulation and analysis in EPR. *J. Magn. Reson.* **178**, 42–55 (2006).

53. Zhang, W. Z. et al. The protein complex crystallography beamline (BL19U1) at the Shanghai Synchrotron Radiation Facility. *Nucl. Sci. Technol.* **30**, 170–180 (2019).

54. Minor, W., Cymborowski, M., Otwinowski, Z. & Chruszcz, M. HKL-3000: the integration of data reduction and structure solution—from diffraction images to an initial model in minutes. *Acta Crystallogr. Sect. D* **62**, 859–866 (2006).

55. Vonrhein, C. et al. Data processing and analysis with the autoPROC toolbox. *Acta Crystallogr. Sect. D* **67**, 293–302 (2011).

56. Adams, P. D. et al. PHENIX: building new software for automated crystallographic structure determination. *Acta Crystallogr. Sect. D* **58**, 1948–1954 (2002).

57. Emsley, P. & Cowtan, K. Coot: model-building tools for molecular graphics. *Acta Crystallogr. Sect. D* **60**, 2126–2132 (2004).

58. Langford, M. L., Zabaleta, J., Ochoa, A. C., Testerman, T. L. & McGee, D. J. In vitro and in vivo complementation of the *Helicobacter pylori* arginase mutant using an intergenic chromosomal site. *Helicobacter* **11**, 477–493 (2006).

59. Jiang, X. et al. The cyclopropane fatty acid synthase mediates antibiotic resistance and gastric colonization of *Helicobacter pylori*. *J. Bacteriol.* **201**, e00374–19 (2019).

60. Akada, J. K. et al. Identification of the urease operon in *Helicobacter pylori* and its control by mRNA decay in response to pH. *Mol. Microbiol.* **36**, 1071–1084 (2000).

## Acknowledgements

This project was supported by grants from the Major Research plan of the National Natural Science Foundation of China (No. 91853118 to Liang Zhang); the National Key R&D Programs of China (No. 2018YFC0311003 to H.B.; 2019YFA0405600 to L.Y.); the National Natural Science Foundation of China (Nos. 22077081, 21722802 to Liang Zhang; 82073899, 31570053, 31870029 to H.B.; 21927814, U1732275, 21825703 to C.T.); Science and Technology Commission of Shanghai Municipality (No. 20S11900300 to Liang Zhang); "Shuguang Program" supported by Shanghai Education Development Foundation and Shanghai Municipal Education Commission (No. 20SG16 to Liang Zhang); Innovative research team of high-level local universities in Shanghai (No. SSMU-ZLCX20180702 to Liang Zhang); The fellowship of China Postdoctoral Science foundation (No. 2020M681342 to Lin Zhang); The National Science Foundation of the Jiangsu Higher Education Institutions of China (No. 18KJA310002 to H.B.); The Jiangsu Specially Appointed Professor and Jiangsu Medical Specialist Programs of China (to H.B.) and Jiangsu Province "Innovative and Entrepreneurial Team" Program (to H.B.) and National Institute of Allergy and Infectious Diseases, (NIH, USA) grant AI15650 to J.E.C.). We thank the staffs from BL19U1 beamline of National Facility for Protein Science in Shanghai (NFPS) at Shanghai Synchrotron Radiation Facility (SSRF), for assistance during data collection. We thank Prof. James Imlay of the University of Illinois for his advice and Prof. Jiahai Zhou of the Shenzhen Institute of Advanced Technology, Chinese Academy of Sciences for providing anaerobic facilities. A portion of this work was performed on the Steady High Magnetic Field Facilities, High Magnetic Field Laboratory, Chinese Academy of Sciences.

## Author contributions

Liang Zhang, H.K.B. and J.E.C. designed experiments; J.S.Z., Lin Zhang, S.Q.S. and P.Z. performed protein purification, kinetics determination, crystallization, and biophysical experiments; J.Y.H. performed the LC–MS/MS analysis; W.Y.S. performed the bioinformatic study; L.P. Zeng, Y.Y.D., Jia Jia and X.D.H. preformed the in vivo and in vitro activity assays; L.Y. and C.L.T. performed the EPR assay; X.X.R., Jing Jiang and Y.N.Z. synthesized chemical probes; H.W.L. and Lu Zhou provided key suggestions during manuscript preparation. Liang Zhang, H.K.B., H.-Z.C. and J.E.C. wrote the paper. All authors discussed and commented on the manuscript.

## Competing interests

The authors declare no competing interests.
