## [Peer Review File · Nature Communications]

Helicobacter pylori FabX contains a [4Fe-4S] cluster essential for unsaturated fatty acid synthesisREVIEWER COMMENTS

Reviewer #1 (Remarks to the Author):

This manuscript describes the structure of a unique dehydrogenase/isomerase from *Helicobacter*. The quality of the work is excellent and the study is clearly presented. The essentiality of FabX is affirmed. Obtaining the structure of the acyl-ACP complex with FabX is also a key advance that expands our understanding of ACP-protein interactions. The crystal structures provide a detailed look at the mechanism. The conclusions are backed by lots of primary and supplementary data. This is a great contribution to the field of bacterial fatty acid biosynthesis. Also, the study contributes to the wider interest in the area of mechanism of FMN- and iron sulfur center-dependent enzymes.

Line 359: 13.4 Å is described as a short distance. It seems like a long distance to this reviewer for electron transfer. Please be more specific here about why 13.4 is the right distance for the process under study.

FabX is about as active with C8 as with C10-ACP. However, C10-ACP is the major substrate in vivo based on 18:1d11 being the major fatty acid. The authors should comment on this apparent disconnect.

Line 501: FabX homologs are mostly found in anaerobic bacteria. FabX requires oxygen to work. Why do anaerobic bacteria have this gene?

Minor points

The PDB submission are preliminary reports and the PDB IDs need to be in the final version.

Ref. 15 could be replaced by the FabM discovery paper (PMID: 12237320)

Line 114: "first natural crystal structure" what is a natural crystal structure.

Fig 2d: there is no black line as stated in the legend from H182

Line 180: please explain what a "reaction trigger" is.

Fig. S3: Is the line a fit of the data to some equation. If so, what equation?

Reviewer #2 (Remarks to the Author):

The paper by Zhou et al. reports a detailed structural and biochemical analysis of the FabX enzyme of *Helicobacter pylori*. They dissected the reaction mechanism of the enzyme in recognizing, regulating and

catalyzing synthesis of cis-3-decenoyl-ACP, the essential intermediate of unsaturated fatty acid (UFA) synthesis. They show that the enzyme contains a [4Fe-4S] electron transfer component, in addition to FMN, which has never been observed in enzymes involved in fatty acid synthesis. This work could be important to design new rational antibacterial compounds targeting *H. pylori*.

This is a strong paper elucidating the catalytic mechanism of an unusual enzyme involved in fatty acid synthesis in bacteria, identifying for the first time a [4Fe-4S] cluster in a bacterial fatty acid biosynthetic enzyme and solving the first crystal structure of a FASII enzyme in complex with acyl-ACP. Nevertheless, the manuscript does not highlight the biological importance that a microaerophilic bacterium synthesizes the cis-3-decenoyl intermediate, a product of the widespread anaerobic pathway, in a reaction that requires oxygen. This issue could be important for microbiologists and the authors should make an effort to transmit the physiological significance of their findings.

In Fig S.1 the authors show the construction of strain BHKS551, but do not describe the physiology of this strain. Is FabA functionally complementing the growth of BHKS551? Is FabA as good as FabX supporting *H. pylori* growth? In a previous paper the same group reported that FabX did not support the growth of a *fabA* *E. coli* strain and that the expression of FabX resulted in low levels of UFAs synthesis. Did the authors determine the fatty acid composition of strain BHKS551? I think the authors should show or comment about these issues since it is pointed out, in page 5, that FabA could access the 3-hydroxydecenoyl-ACP intermediate to produce cis-3-decenoyl fatty acids in *H. pylori*, that I assume are converted to cyclopropane fatty acids.

Reviewer #3 (Remarks to the Author):

In this manuscript, the authors report an insightful structural and catalytic characterization of the fatty acid dehydrogenase, FabX, from *H. pylori*. Their study shows that FabX is a unique oxygen dependent dehydrogenase that requires FMN and a [4Fe4S] cluster as cofactors to function properly. The crystal structure of this protein reveals that both cofactors are within the electron-transfer distance and that they are connected by a kinked, "L" shaped substrate channel. In addition, two holo-enzyme structures – one with the native decenoyl fatty acid substrate bound and the other with an artificial octanoyl fatty acid substrate bound – demonstrate that the fatty acid substrates are bound within the "L" shaped channel, in which the atom of hydrogen abstraction is aligned with the oxyanion hole observed in the

apo enzyme structure. In parallel to crystallographic analyses, EPR investigations provide insights into the electron transfer process, revealing a signal at 120K that originates from FMNH• in the presence of lower equivalents of dithionite and another signal at 10K that originates from [4Fe4S]1+ in the presence of higher equivalents of dithionite, as well as similar signals when the enzyme is treated with substrates. Moreover, spin trap DMPO and colorimetric assays demonstrate the formation of oxygen radicals and hydrogen peroxide, respectively, during catalysis. Based on these results, the authors propose a reaction cycle that involves hydrogen abstraction by FMN, generation of FMNH•, and reduction of [4Fe4S]2+ to [4Fe4S]1+. Two molecules of oxygen are proposed to come in after the release of product, resulting in the formation of oxygen radicals upon removal of two electrons, which recycles the enzyme back to the starting oxidized state.

This study features a unique oxygen dependent dehydrogenase that requires a [4Fe4S] cluster as an electron reservoir. This observation is unusual given that [4Fe4S] clusters are typically unstable in the presence of oxygen, and other similar enzymes often unload two electrons onto the oxygen molecules that enter the reaction site successively during catalysis. In the case of FabX, the authors argue on the basis of the holo-enzyme structure that the entry of oxygen molecules may be blocked by the presence of substrate or product, and thus the [4Fe4S] cluster is required to serve as a temporary holding place for the extra electron. This finding sheds important light on how FabX produces a large amount of ROS and hydrogen peroxide, which could bear significant relevance to the pathogenicity of H pylori. It also provides an interesting new concept of the mechanism of oxygenase, which splits the two electrons derived from hydrogen abstraction between two different cofactors. Because of the novelty and importance of this work, I recommend acceptance of this manuscript for publication once the following concerns are properly addressed.

1. Pg 3, paragraph 1: The authors' statement that "the hydrophobic imidazole ring of H182 sidechain is required for orienting FMN by 198 hydrophobic interactions" seems to contradict their latter suggestion of a "loss of the π - π stacking interactions between H182 imidazole ring and FMN". Given that the H to F mutation of the enzyme retains most of the FMN cofactor, π stacking seems to be a better explanation here. The statement regarding hydrophobic interaction should either be removed or revised.

2. The EPR experiment of the enzyme titrated with dithionite is somewhat problematic (Fig 5a, 5b). It is known that dithionite can decompose into various sulfurous species that release different amounts of electrons. Therefore, it is inaccurate to assign one equivalent of dithionite as one equivalent of electron. In addition, decomposition of dithionite results in different solution potentials at different concentrations and/or pH values, making it even more difficult to interpret the results. To overcome this problem, the EPR/titration experiments should be performed with a more well-defined reductant, such as MV; alternatively, the titration experiment could be done in the reverse direction, i.e., starting with the dithionite-reduced enzyme, followed by oxidation with ferricyanide or other suitable oxidants.

3. Fig 5c-e: The reaction conditions of the EPR experiments are unclear in terms of whether oxygen was introduced. Given the presence of ROS, it seems that oxygen is present in the spin trap experiment (Fig 5e). Is oxygen also present in the other experiment? This would explain why the [4Fe4S]₁₊ signal appears to be smaller than those in Fig 5a.

4. The presence of oxygen and ROS seems detrimental for the integrity of the [4Fe4S] cluster. Can the authors comment on that on the basis of the crystal structure?

Minor points:

1. Pg 8, line 219: Remove “right” from the sentence “... suggests that the [4Fe4S] binding domain in FabX right replaces the NAD(P)H binding motif of PgFabK...”.

2. Fig 5c, d: The g values of the EPR signals should be indicated.

上海交通大学

SHANGHAI JIAO TONG UNIVERSITY

Point-by-point response

Manuscript ID: NCOMMS-21-27596

Referee #1:

This manuscript describes the structure of a unique dehydrogenase/isomerase from *Helicobacter*. The quality of the work is excellent and the study is clearly presented. The essentiality of fabX is affirmed. Obtaining the structure of the acyl-ACP complex with FabX is also a key advance that expands our understanding of ACP-protein interactions. The crystal structures provide a detailed look at the mechanism. The conclusions are backed by lots of primary and supplementary data. This is a great contribution to the field of bacterial fatty acid biosynthesis. Also, the study contributes to the wider interest in the area of mechanism of FMN- and iron sulfur center-dependent enzymes.

Question 1: Line 359: 13.4 Å is described as a short distance. It seems like a long distance to this reviewer for electron transfer. Please be more specific here about why 13.4 is the right distance for the process under study.

Response: We thank the reviewer for the comments. The conclusion of 13.4 Å distance came from the reference published in 1999 (Nature, 1999, 402, 47). In the reference, they analyzed the enzymes with known atomic structure deposited in the Protein Data Bank (PDB), whose function involves electron transfer. They found that electrons can travel up to 14 Å between redox centers through the protein medium (in the reference paper Fig. 1c), and by favoring proteins with redox centers placed within 14 Å of each other, natural selection fosters robust electron-transfer designs (in the reference paper Fig. 1d). In FabX, the distance between FMN and the [4Fe-4S] cluster is 13.4 Å, and thereby we believe that the electrons transfer directly between the FMN and the [4Fe-4S] cluster. We have revised our manuscript accordingly (page 17, discussion section, first paragraph, lines 9-14).

Question 2: FabX is about as active in dehydrogenation of C8 as with C10-ACP. However, C10-ACP is the major substrate *in vivo* based on 18:1d11 being the major fatty acid. The authors should comment on this apparent disconnect.

Response: Thank the reviewer for the comments. Our enzymatic results have confirmed that C8-ACP is the dehydrogenation substrate of FabX, although the enzymatic turnover of FabX in catalyzing C8-ACP is two-folds slower than that of C10-ACP (Table S1). This is indeed a known characteristic of FAS-II enzymes. Previous studies demonstrated that FabA, the dehydratase/isomerase that catalyzes unsaturated fatty acid biosynthesis in most of the bacteria, performs dehydration of C8 and C12 substrates about 80% as well as C10 *in vitro* (J. Biol. Chem., 1996, 271, 27795). Moreover, this enzymatic difference could be explained by the structural comparison and docking of C10 substrate to FabX structure (Fig. S26). Since there is not enough space inside the active pocket tunnel to accommodate the C10-acyl substrate, there must be some conformational changes of FabX around the active pocket tunnel occur somehow to overcome this issue, and thereby increase the catalytic turnover in catalyzing C10-ACP. Moreover, we have previously demonstrated that the fatty acid dehydratase FabZ, the homolog of FabA, occurs large conformational changes around the active pocket tunnel to enlarge the space of the active pocket tunnel for long acyl chain substrate accommodation (Cell Res. 2016, 26, 1330). The conformational changes

also facilitate ACP leaving after FabZ catalysis in a see-saw manner, and thereby increases its catalytic turnover on long acyl chain substrates. Hence, these results suggested that conformational changes in FAS-II enzymes upon the binding of long acyl substrate is a comprehensive process. We have revised our manuscript accordingly (page 13, lines 6-17).

Question 3: Line 501: FabX homologs are mostly found in anaerobic bacteria. FabX requires oxygen to work. Why to anaerobic bacteria have this gene?

Response: Thanks for the comments. Our bioinformatic results suggested that FabX homologs predominantly exist in microaerophilic bacteria rather than completely anaerobic bacteria, such as in bacteria living in marine, hydrothermal springs, human oral cavity, and stomach (Fig. S30). These environments are not completely anaerobic, and contain low concentrations of oxygen at the level that would not destroy the cluster but provide the required oxygen as the electron acceptor. In *H. pylori* case, we demonstrated that the oxygen is not only utilized to synthesize unsaturated fatty acids. More importantly, the ROS product of the FabX catalysis support its pathogenic function in the corrosion of gastric mucosa. Given this finding, we believe that other FabX dependent microaerophilic bacterium may also utilize the product from oxygen to play species-specific roles which remain unknown and thereby to support their living in their extreme environment. We have revised the manuscript accordingly (Page 18, discussion section, lines 9-18).

Question 4: Minor points:

The PDB submission are preliminary reports and the PDB IDs need to be in the final version.

Response: Thanks for the comment. We have updated the structural reports to the final version.

Question 5: Ref. 15 could be replaced by the FabM discovery paper (PMID: 12237320)

Response: Thanks for the comment. We have revised the reference #15 to FabM paper.

Question 6: Line 114: “first natural crystal structure”; what is a natural crystal structure.

Response: Thanks for the comment. We have revised the sentence to “this is the first non-crosslinked crystal structure of a FAS-II enzyme in complex with acyl-ACP”.

Question 7: Fig 2d: there is no black line as stated in the legend from H182

Response: Thanks for the comment. We have removed the sentence from the legend.

Question 8: Line 180: please explain what at “reaction trigger” is.

Response: Thanks for the comment. We have revised the sentence to “previous studies on NMO family enzymes such as FabKs suggested that such catalytic histidine functions as a “reaction trigger” as the N^{δ1} atom of the sidechain of the histidine residue abstract a proton from the substrate through nucleophilic attack to initiate the reaction, and facilitate the subsequent electron transfer on FMN”.

Question 9: Fig. S3: Is the line a fit of the data to some equation. If so, what equation?

Response: Thanks for the comment. The curve was fitted to the Michaelis-Menten equation. We have

revised the figure legend accordingly.

Referee #2:

The paper by Zhou et al. reports a detailed structural and biochemical analysis of the FabX enzyme of *Helicobacter pylori*. They dissected the reaction mechanism of the enzyme in recognizing, regulating and catalyzing synthesis of *cis*-3-decenoyl-ACP, the essential intermediate of unsaturated fatty acid (UFA) synthesis. They show that the enzyme contains a [4Fe-4S] electron transfer component, in addition to FMN, which has never been observed in enzymes involved in fatty acid synthesis. This work could be important to design new rational antibacterial compounds targeting *H. pylori*.

This is a strong paper elucidating the catalytic mechanism of an unusual enzyme involved in fatty acid synthesis in bacteria, identifying for the first time a [4Fe-4S] cluster in a bacterial fatty acid biosynthetic enzyme and solving the first crystal structure of a FASII enzyme in complex with acyl-ACP.

Question 1: Nevertheless, the manuscript does not highlight the biological importance that a microaerophilic bacterium synthesizes the *cis*-3-decenoyl intermediate, a product of the widespread anaerobic pathway, in a reaction that requires oxygen. This issue could be important for microbiologists and the authors should make an effort to transmit the physiological significance of their findings.

Response: We thank the reviewer for the comments. The finding that the dehydrogenase activity of FabX requires oxygen has been shown in our previous paper (Fig. 5, Cell Chem Biol. 2016, 23, 1480). We have demonstrated that the degassing FabX reactions or anaerobic condition inactivates the dehydrogenation reaction. Our bioinformatics data (Fig. S30 in this manuscript) has shown that FabX and its homologs predominantly exist in microaerophilic bacterium rather than anaerobes. Our catalytic results provide an understanding of how microaerophilic bacterium utilizes the small amount of oxygen from the environment. In *H. pylori* case, we demonstrated that the oxygen is not only utilized to synthesize unsaturated fatty acids. More importantly, the ROS product from the FabX catalysis supports its pathogenic function in the corrosion of gastric mucosa. Given this finding, we believe that other FabX-homolog dependent microaerophilic bacterium may also utilize the product from oxygen to play species-specific roles which remain unknown, and thereby to support their living in their extreme environments. We have revised the manuscript accordingly (Page 18, discussion section, lines 9-18).

Question 2: In Fig S.1 the authors show the construction of strain BHKS551, but do not describe the physiology of this strain. Is FabA functionally complementing the growth of BHKS551? Is FabA as good as FabX supporting *H. pylori* growth? In a previous paper the same group reported that FabX did not support the growth of a *fabA E. coli* strain and that the expression of FabX resulted in low levels of UFAs synthesis. Did the authors determine the fatty acid composition of strain BHKS551? I think the authors should show or comment about these issues since it is pointed out, in page 5, that FabA could access the 3-hydroxydecenoyl-ACP intermediate to produce *cis*-3-decenoyl fatty acids in *H. pylori*, that I assume are converted to cyclopropane fatty acids.

Response: We thank the reviewer for the comments. FabA is a bifunctional enzyme with dehydratase and isomerase activity. In *E. coli*, FabA catalyzes the dehydration of 3-OH-decenoyl-ACP to

上海交通大学

SHANGHAI JIAO TONG UNIVERSITY

trans-2-decenoyl-ACP, then the isomerization of the latter to *cis*-3-decenoyl-ACP, which is the key step of the classical unsaturated fatty acid biosynthetic pathway. Due to the essential role of *fabX* in *H. pylori*, we introduced a synthetic copy of the *E. coli fabA* gene to allow survival of *H. pylori* strains lacking the *fabX* gene. Note that during the construction of strain BHKS551 (NSH57 IR0203::*fabA* Δ *fabX*), the synthetic DNA fragment containing the *fabA* gene with a *H. pylori* promoter (the *ureI* promoter) was inserted into the *H. pylori* genome. Additionally, the *fabA* gene is optimized for *H. pylori* codons to make sure it is expressed well in *H. pylori*. To response to the reviewer's concern, we have determined the growth phenotype in brain heart infusion (BHI) broth (**Fig. S2**). The growth curves of strains BHKS551 (NSH57 IR0203::*fabA* Δ *fabX*) and BHKS568 (NSH57 IR0203::*fabX* Δ *fabX*) showed that *EcFabA* functionally replaces *FabX* that supports *H. pylori* growth. This result is different from our previous paper (Cell Chem Biol. 2016) showing that *FabX* did not support the growth of a *E. coli fabA* knockout strain due to low levels of UFAs synthesis by *FabX*. First, the two host strains, *E. coli* and *H. pylori* are different in that they may require different levels of UFAs to support growth. Second, *FabX* may not efficiently recognize *E. coli* ACP, whereas *FabA* could recognize *H. pylori* ACP. Third, the ability of FAS (fatty acid biosynthesis) enzymes and ACP to complement loss of *E. coli* ACP varies from bacterium to bacterium. For example, *Enterococcus faecalis* ACP does not replace *E. coli* ACP but FAS enzymes function in *E. coli* (J. Biol. Chem., 2007, 282,20319; J. Biol. Chem., 2004, 279, 34489). Since this section is intended to show the essential role of *fabX* in *H. pylori*, we did not determine the fatty acid composition of strain BHKS551. We have revised the manuscript accordingly (Page 6, lines 4-5; supplementary Fig. S2).

Referee #3:

In this manuscript, the authors report an insightful structural and catalytic characterization of the fatty acid dehydrogenase, *FabX*, from *H. pylori*. Their study shows that *FabX* is a unique oxygen dependent dehydrogenase that requires FMN and a [4Fe4S] cluster as cofactors to function properly. The crystal structure of this protein reveals that both cofactors are within the electron-transfer distance and that they are connected by a kinked, “L” shaped substrate channel. In addition, two holo-enzyme structures – one with the native decenoyl fatty acid substrate bound and the other with an artificial octanoyl fatty acid substrate bound – demonstrate that the fatty acid substrates are bound within the “L” shaped channel, in which the atom of hydrogen abstraction is aligned with the oxyanion hole observed in the apo enzyme structure. In parallel to crystallographic analyses, EPR investigations provide insights into the electron transfer process, revealing a signal at 120K that originates from FMNH• in the presence of lower equivalents of dithionite and another signal at 10K that originates from [4Fe4S]1+ in the presence of higher equivalents of dithionite, as well as similar signals when the enzyme is treated with substrates. Moreover, spin trap DMPO and colorimetric assays demonstrate the formation of oxygen radicals and hydrogen peroxide, respectively, during catalysis. Based on these results, the authors propose a reaction cycle that involves hydrogen abstraction by FMN, generation of FMNH•, and reduction of [4Fe4S]2+ to [4Fe4S]1+. Two molecules of oxygen are proposed to come in after the release of product, resulting in the formation of oxygen radicals upon removal of two electrons, which recycles the enzyme back to the starting oxidized state.

This study features a unique oxygen dependent dehydrogenase that requires a [4Fe4S] cluster as an electron reservoir. This observation is unusual given that [4Fe4S] clusters are typically unstable in the presence of oxygen, and other similar enzymes often unload two electrons onto the oxygen molecules that

上海交通大学

SHANGHAI JIAO TONG UNIVERSITY

enter the reaction site successively during catalysis. In the case of FabX, the authors argue on the basis of the holo-enzyme structure that the entry of oxygen molecules may be blocked by the presence of substrate or product, and thus the [4Fe4S] cluster is required to serve as a temporary holding place for the extra electron. This finding sheds important light on how FabX produces a large amount of ROS and hydrogen peroxide, which could bear significant relevance to the pathogenicity of *H. pylori*. It also provides an interesting new concept of the mechanism of oxygenase, which splits the two electrons derived from hydrogen abstraction between two different cofactors. Because of the novelty and importance of this work, I recommend acceptance of this manuscript for publication once the following concerns are properly addressed.

Question 1. Pg 3, paragraph 1: The authors' statement that "the hydrophobic imidazole ring of H182 sidechain is required for orienting FMN by 198 hydrophobic interactions" seems to contradict their latter suggestion of a "loss of the π - π stacking interactions between H182 imidazole ring and FMN". Given that the H to F mutation of the enzyme retains most of the FMN cofactor, π stacking seems to be a better explanation here. The statement regarding hydrophobic interaction should either be removed or revised.

Response: Thank for the comments. we revised the sentences to "Substitution of H182 with the nearly isosteric glutamine (FabX H182Q) resulted in a sharp decrease of the FMN content to ~28%, reduced protein stability by 7.2°C, and completely abolished enzymatic activity. These observations suggested that the imidazole ring of H182 is required for maintaining both protein stability and the FMN cofactor for achieving dehydrogenation activity due to π stacking" (Page 8, lines 5-10).

Question 2 The EPR experiment of the enzyme titrated with dithionite is somewhat problematic (Fig 5a, 5b). It is known that dithionite can decompose into various sulfurous species that release different amounts of electrons. Therefore, it is inaccurate to assign one equivalent of dithionite as one equivalent of electron. In addition, decomposition of dithionite results in different solution potentials at different concentrations and/or pH values, making it even more difficult to interpret the results. To overcome this problem, the EPR/titration experiments should be performed with a more well-defined reductant, such as MV; alternatively, the titration experiment could be done in the reverse direction, i.e., starting with the dithionite-reduced enzyme, followed by oxidation with ferricyanide or other suitable oxidants.

Response: We thank the reviewer for the comments on the EPR experiment. Suggested by the referee, the EPR titration experiment was repeated and was further improved.

Although the reductant MV (methyl viologen) is indeed used more widely in redox titration experiments, it is a two-electron carrier and undergoes a radical cation ($MV^{•+}$) state during the oxidation from the reduced state (neutral MV^0) to the oxidized state (bivalent cation MV^{2+}). Thus, MV could not be used in the titration of FabX because its radical cation ($MV^{•+}$) would give rise to EPR signal (*J. Mag. Reson.*, 1980, 40, 351; *Faraday Discuss.*, 2021, 225, 414), which would interfere the detection and quantification of the FMNH• radical during titration.

Instead, we took the other advice offered by the reviewer, and performed oxidative titration (Please see the figure below). Briefly, FabX was first anaerobically reduced with freshly prepared sodium dithionite and excess dithionite was removed using desalting column. The reduced protein was then progressively reoxidized by adding increasing molar equivalents (with respect to protein) of ferricyanide ($K_3Fe(CN)_6$), while the signal of reduced [4Fe-4S] $^{1+}$ and FMNH• radical were monitored using EPR spectroscopy. EPR

上海交通大学

SHANGHAI JIAO TONG UNIVERSITY

spectrum was collected at 10K to highlight signal from $[4\text{Fe-4S}]^{1+}$, and at 120K to highlight signal from FMNH• radical. To our delight, oxidative titration gave rise to EPR spectrum in reverse order compared to those from reductive titration using dithionite. Specially, only EPR signal from $[4\text{Fe-4S}]^{1+}$ was observed in the fully-reduced state (black trace), indicating that the $[4\text{Fe-4S}]$ cluster and FMN were both reduced to $[4\text{Fe-4S}]^{1+}$ and FMNH₂. When one molar equivalent (with respect to protein) of ferricyanide was added, signal from both $[4\text{Fe-4S}]^{1+}$ and FMNH• radical was observed (red trace), indicating that ferricyanide took away the first electron from FMNH₂, resulting in the formation of FMNH• radical. The addition of two molar equivalents of ferricyanide almost abolished EPR signal from $[4\text{Fe-4S}]^{1+}$ and left only the EPR signal from FMNH• radical (blue trace), implying that ferricyanide accept the second electron from the $[4\text{Fe-4S}]$ cluster. $[4\text{Fe-4S}]^{1+}$ cluster was oxidized to EPR-silent $[4\text{Fe-4S}]^{2+}$ state. Lastly, all EPR signal was lost upon the addition of the third equivalent of ferricyanide (magenta trace), indicating the protein was fully oxidized. In this state, $[4\text{Fe-4S}]^{1+}$ cluster and FMNH₂ was oxidized to EPR-silent $[4\text{Fe-4S}]^{2+}$ and FMN, respectively. In this way, the conclusion from titration experiment using dithionite was further confirmed by the oxidative titration. Specifically, the re-oxidative titration reached two conclusions: (i) one reduced FabX molecule can donate up to three electrons to oxidant and become fully oxidized; (ii) The reduced FMNH₂ preferentially loses the first electron to oxidant and becomes FMNH• radical; reduced $[4\text{Fe-4S}]^{1+}$ cluster donates the second electron and FMNH• radical donates the third electron.

In short, our oxidative titration further confirmed the findings from the reductive titration in our manuscript. Moreover, to our knowledge, dithionite has been used as reductive titrants in studies of multicopper oxidases and P450_{BM-3} (*J. Am. Chem. Soc.*, 2015, 137, 8783; *Arch. Biochem. Biophys.*, 1992, 294, 654). Given these results, Figure 5 has now been updated using “electron equivalents of dithionite” instead of “dithionite”; The description of the titration experiment in main text, figure legend and supporting information has been updated accordingly; The result of oxidative titration is added to the supporting information (Fig. S27).

Figure S27. Oxidative titration of reduced FabX, monitored by EPR. Different molar equivalents

(with respect to protein) of ferricyanide($K_3Fe(CN)_6$) were titrated to dithionite-reduced FabX and EPR spectra were collected using same instrumental parameters.

Question 3 Fig 5c-e: The reaction conditions of the EPR experiments are unclear in terms of whether oxygen was introduced. Given the presence of ROS, it seems that oxygen is present in the spin trap experiment (Fig 5e). Is oxygen also present in the other experiment? This would explain why the $[4Fe_4S]^{1+}$ signal appears to be smaller than those in Fig 5a.

Response: We apologize for the unclear description of reaction conditions. Yes, oxygen was present for both the spin-trapping EPR experiment (Fig. 5e) and the reaction of FabX with decanoyl-ACP (Fig. 5c, 5d). The experimental details have been added to the figure legend and supporting information accordingly.

Question 4: The presence of oxygen and ROS seems detrimental for the integrity of the $[4Fe_4S]$ cluster. Can the authors comment on that on the basis of the crystal structure?

Response: Thank for the comment. We have revised our manuscript to “During dehydrogenation catalysis by FabX, oxygen is required as the final electron acceptor. However, the maintenance of the reduced status of the $[4Fe_4S]$ cluster is required for its activity during electron transfer. High concentrations of oxygen or the superoxide/ROS products inside the cell are detrimental for its integrity as the cluster is located very close to the protein surface. Hence, on one hand, FabX homologs mostly exist in microaerophilic bacteria to avoid a high concentration of oxygen in the environment. On the other hand, *H. pylori* utilizes the low concentration of oxygen to generate superoxide/ROS and excretes it to extracellular environment to protect FabX from over-oxidation and achieve its pathogenic function in the corrosion of gastric mucosa.” (page 18, discussion section, first paragraph, lines 10-19).

Question 5: minor points:

Pg 8, line 219: Remove “right” from the sentence “... suggests that the $[4Fe_4S]$ binding domain in FabX right replaces the NAD(P)H binding motif of PgFabK...”.

Response: Thank for the comment. We have removed it from the sentence.

Question 6: Fig 5c, d: The g values of the EPR signals should be indicated.

Response: Thanks for the suggestion. The g values for EPR peaks corresponding to $[4Fe_4S]^{1+}$ and $FMNH\cdot$ radical have been added to Fig. 5c and 5d.

REVIEWERS' COMMENTS

Reviewer #1 (Remarks to the Author):

None.

Reviewer #2 (Remarks to the Author):

My concerns have been addressed in the revision of the manuscript. This is an important contribution to the field of bacteria fatty acid biosynthesis

Reviewer #3 (Remarks to the Author):

The authors have addressed all my concerns and this nice report is now ready for publication.

上海交通大学

SHANGHAI JIAO TONG UNIVERSITY

Point-by-point response

Manuscript ID: NCOMMS-21-27596

Referee #1:

None.

Referee #2:

My concerns have been addressed in the revision of the manuscript. This is an important contribution to the field of bacteria fatty acid biosynthesis. This is a strong paper elucidating the catalytic mechanism of an unusual enzyme involved in fatty acid synthesis in bacteria, identifying for the first time a [4Fe-4S] cluster in a bacterial fatty acid biosynthetic enzyme and solving the first crystal structure of a FASII enzyme in complex with acyl-ACP.

Response: We thank the reviewer for the positive comments.

Referee #3:

The authors have addressed all my concerns and this nice report is now ready for publication.

Response: We thank the reviewer for the positive comments.